# XYZCylinder: Feedforward Reconstruction for Driving Scenes Based on a Unified Cylinder Lifting Method

## Abstract

Recently, more attention has been paid to feedforward reconstruction paradigms, which mainly learn a fixed view transformation implicitly and reconstruct the scene with a single representation. However, their generalization capability and reconstruction accuracy are still limited while reconstructing driving scenes, which results from two aspects: (1) The fixed view transformation fails when the camera configuration changes, limiting the generalization capability across different driving scenes equipped with different camera configurations. (2) The small overlapping regions between sparse views of the $360°$ panorama and the complexity of driving scenes increase the learning difficulty, reducing the reconstruction accuracy. To handle these difficulties, we propose **XYZCylinder**, a feedforward model based on a unified cylinder lifting method which involves camera modeling and feature lifting. Specifically, to improve the generalization capability, we design a Unified Cylinder Camera Modeling (UCCM) strategy, which avoids the learning of viewpoint-dependent spatial correspondence and unifies different camera configurations with adjustable parameters. To improve the reconstruction accuracy, we propose a hybrid representation with several dedicated modules based on newly designed Cylinder Plane Feature Group (CPFG) to lift 2D features to 3D space. Extensive experimental results show that XYZCylinder achieves state-of-the-art performance under different evaluation settings, and can be generalized to other driving scenes in a zero-shot manner. Anonymous project page: here.

## 1 Introduction

3D reconstruction focuses on building a 3D digital model with spatial structure and visual fidelity from the limited views of 2D images, which has been a hot topic in computer graphics and computer vision, and has been widely used in many tasks, for example, autonomous driving. In this paper, we focus on the reconstruction of driving scenes using the sparse views of one timestamp.

Previous iterative reconstruction methods, while capable of high accuracy, are too computationally expensive for large-scale 3D asset collection. Generally, these iterative methods (Gao et al., 2024a; Liu et al., 2024a; Yu et al., 2024; Mihajlovic et al., 2024) reconstruct the scene based on NeRF (Mildenhall et al., 2021) or 3DGS (Kerbl et al., 2023) representations, which need to be iteratively optimized for different scenes. The inherent high latency and computational cost of the optimization process hinder iterative reconstruction methods from being applicable to efficiency-intensive tasks. In contrast, feedforward reconstruction methods (Tian et al., 2025; Wei et al., 2025; Gieruc et al., 2024) reconstruct the scenes within a single forward pass and generalize to different scenes, making them attract more and more attention from researchers.

Existing feedforward reconstruction methods (Zhang et al., 2025; Charatan et al., 2024; Chen et al., 2024; Xu et al., 2025; Liu et al., 2025b; Min et al., 2024; Wang et al., 2024b;a; Tang et al., 2025; Fei et al., 2024a) mainly learn a fixed view transformation implicitly and reconstruct the scene with a single representation. Their generalization capability and reconstruction accuracy are limited to some extent when reconstructing driving scenes. For example, different cars may have different camera configurations, including the number of cameras and the extrinsic and intrinsic parameters of the camera. The view-dependent design learns the fixed view transformation, limiting their

generalization for different driving scenes. When meeting various camera configurations, they suffer from severe quality degradation, requiring a complete redesign and re-training of the networks. Besides, their reconstruction capability relies on the pixel-based spatial correspondence learned from the highly overlapping regions between different views, which may fail when the overlapping regions become small (*e.g.*, ∼6 views for 360° panorama in common driving scenes). In addition, these methods produce the "2.5D" scene representation based on per-pixel depths and offsets, which suffers from the issues of deformation, holes, and distortion due to the scale ambiguity and incomplete geometry when meeting complex driving scenes. Though some methods (Wei et al., 2025; Gieruc et al., 2024) reconstruct driving scenes with compact cross-view 3D representations (*i.e.*, Triplane (Chan et al., 2022) and Tri-Perspective View (TPV) (Huang et al., 2023)), they suffer from performance degradation with large-angle view transformation and sparse geometric cues.

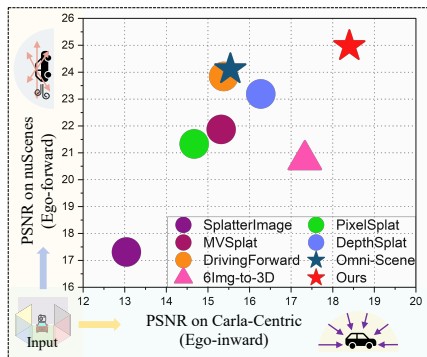

Figure 1: Reconstruction results of the proposed method under different evaluation settings.

To handle these issues, we propose **XYZCylinder**, a feedforward reconstruction method for driving scenes based on a unified cylinder lifting method. This method includes a camera modeling strategy and feature groups. (1) To improve the generalization capability, we design a Unified Cylinder Camera Modeling (UCCM) strategy (Section 3.1), where the learning-based view transformation is replaced by a deterministic and explicit mapping, omitting the learning of spatial correspondence. By adjusting the training-free parameters during the construction of the cylinder planes, the proposed method can achieve zero-shot reconstruction with different camera configurations, showing the generalization capability of XYZCylinder. (2) To improve the reconstruction accuracy, we formulate the scene reconstruction as a hybrid representation decoded by the newly designed Cylinder Plane Feature Group (CPFG). With the dedicated occupancy-aware, volume-aware, and pixel-aware modules, CPFG lifts the features from 2D space to 3D space (Section 3.2). A background-foreground decoupling strategy is also adopted for better modeling (Section 3.3). To show the effectiveness of the proposed XYZCylinder, both ego-forward Wei et al. (2025) and ego-inward (Gieruc et al., 2024) evaluation settings are adopted, as shown in Fig. 1. Extensive experimental results show that XYZCylinder achieves state-of-the-art reconstruction results across different evaluation settings, and can be generalized well to other driving scenes with different camera configurations in a zero-shot manner. The main contributions in this paper include:

- We design a unified cylinder camera modeling strategy to map different views to a unified cylinder plane with adjustable parameters, improving the generalization capability of XYZCylinder.
- We model the driving scene using a hybrid representation with dedicated modules based on the cylinder plane feature group, enhancing the reconstruction accuracy of XYZCylinder.
- The proposed XYZCylinder achieves state-of-the-art reconstruction results under different evaluation settings and can also be generalized to different driving scenes in a zero-shot manner.

## 2 RELATED WORK

**Feedforward Reconstruction Models.** Feedforward reconstruction methods are defined by their scene representation. Volumetric methods (Gieruc et al., 2024; Ren et al., 2024; Zou et al., 2024) ensure geometric completeness and handle occlusions effectively, yet they often produce overly smooth surfaces lacking high-frequency detail. Conversely, pixel-based models (Wang et al., 2024a; Tang et al., 2025; Wang et al., 2025a; Charatan et al., 2024; Chen et al., 2024; Xu et al., 2025; Zhang et al., 2025; Fei et al., 2024a; Miao et al., 2025; Liu et al., 2024b; Zheng et al., 2025; Fei et al., 2024b; Wang et al., 2024b; 2025b; Liu et al., 2025b; Xiao et al., 2025; Li et al., 2024b; Min et al., 2024; Smart et al., 2024) excel at generating dense geometry in forward-facing scenarios but suffer from voids and occlusions in outward-facing settings like autonomous driving. This trade-off has motivated hybrid methods like Omni-Scene (Wei et al., 2025). Our work advances this paradigm, distinguishing itself from Omni-Scene's pixel-guided strategy by employing four prediction branches for each representation, which yields a more complete and detailed geometric reconstruction.

Figure 2: **Overview of XYZCylinder.** The scene is reconstructed in three stages with the unified cylinder camera modeling for feature extraction, and a hybrid representation with different dedicated modules for foreground and background reconstruction.

**2D Feature Lifting.** Lifting 2D features to a 3D representation is fundamental to 3D perception. CaDDN (Reading et al., 2021) projects 2D features into a 3D volume using a predicted categorical depth distribution. LSS (Philion & Fidler, 2020) forms a Bird's-Eye-View (BEV) map by projecting view frustum features with a predicted per-pixel depth distribution. BEVFormer (Li et al., 2024c) employs spatial cross-attention to convert multi-camera 2D features into a unified BEV representation. TPVFormer (Huang et al., 2023) uses a transformer to construct three orthogonal planes from 2D inputs. DFA3D (Li et al., 2023) integrates depth with a 3D deformable attention mechanism to lift 2D features. In contrast, our approach leverages feature spatial consistency and channel reordering to restructure 2D features into a pillar-based field for subsequent 3D tasks. A more detailed review of related work is provided in Appendix A.3.

## 3 METHODOLOGY

Given $N$ views (*e.g.*, $N = 6$ in common driving scenes) in one timestamp, the goal is to reconstruct the corresponding 3D scene. Formally, the $n$-th view is denoted by a tuple $\mathcal{C}_n = \{\mathbf{I}_n, f_n, \mathbf{P}_n, \mathbf{W}_n^e, \mathbf{c}_n^e\}$, where $\mathbf{I}_n \in \mathbb{R}^{3 \times H \times W}$ is the image captured with the vertical Field of View (FoV) $f_n$, and $\mathbf{P}_n \in \mathbb{R}^{4 \times 4}$, $\mathbf{W}_n^e \in \mathbb{R}^{4 \times 4}$, and $\mathbf{c}_n^e \in \mathbb{R}^3$ are the intrinsic parameters of the camera, extrinsic parameters mapping from the camera to the ego coordinate system, and the position of the camera in the ego-vehicle coordinate system, respectively. As shown in Fig. 2, the reconstruction pipeline is divided into three stages: (1) In the feature extraction stage (Section 3.1), a Unified Cylinder Camera Modeling (UCCM) is designed to project different views into a unified cylinder plane; (2) In the foreground reconstruction stage (Section 3.2), the Occupancy-Aware Module, Volume-Aware Module, and Pixel-Aware Module are designed to produce Cylinder Plane Feature Groups (CPFGs); (3) In the background reconstruction and fusion stage (Section 3.3), the background (*i.e.*, sky and clouds) is generated in the 2D space and fused with the rendered 2D foreground image.

### 3.1 FEATURE EXTRACTION

Processing the 360° field of view from $N$-surrounded cameras is nontrivial. To do this, we design a Unified Cylinder Camera Modeling (UCCM) strategy to efficiently project these $N$ views into a cylinder plane as shown in Fig. 3. UCCM defines a cylindrical coordinate system using five key parameters: the coordinates of the central point $\mathbf{c}_u^x \in \mathbb{R}^3$, radius $R_u \in \mathbb{R}$, height $Z_u \in \mathbb{R}$, and resolution $H_u \times W_u$ of the cylinder plane.

**Construction of Cylinder Plane.** The central point $\mathbf{c}_u^x$ is computed by averaging the 3D positions of the $N$ cameras with a small height offset $\mathbf{c}_u^x = \sum_{n=0}^{N-1} \mathbf{c}_n^e / N + (0, 0, \Delta h)^T$, while the cylinder's height $Z_u$ is derived from the vertical extent of the scene. Specifically, given a voxelized occupancy grid $\mathbf{O} \in \mathbb{R}^{L_o \times H_o \times W_o}$, we first extract its corresponding point cloud $\mathbf{X}_o \in \mathbb{R}^{3 \times L_o \times H_o \times W_o}$. The height $Z_u$ is the distance between the top-most and bottom-most points along the vertical axis.

In an ideal scenario where the $N$ cameras share the same location and intrinsic parameters, and only differ in the poses, the FoV would perfectly match the vertical angle required for cylindrical projection. Unfortunately, there exist inevitable camera-to-camera and camera-to-central axis off-

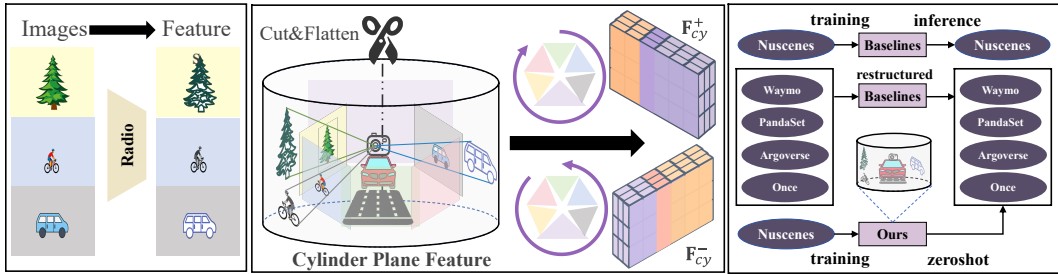

Figure 3: **Overview of the unified cylinder camera modeling (UCCM).** The design of UCCM empowers our model with zero-shot generalization across different datasets.

sets, leaving the top and bottom regions in the cylinder plane unprojected and introducing boundary artifacts. To mitigate this issue, the effective vertical angle is set to the minimum FoV $f_{min} \in \mathbb{R}$ of these $N$ views, which is further multiplied by a factor of $\rho$, thereby mitigating the unprojected regions. Then, the radius is calculated by $R_u = \frac{Z_u/2}{\tan(\rho \cdot f_{min}/2)}$.

**Production of Cylinder Plane Feature.** Given the $n$-th view, the image $\mathbf{I}_n$ is fed into the feature extractor (*i.e.*, Radio-v2.5 (Heinrich et al., 2024)) to produce the image feature $\mathbf{F}_n$. Then we need to get the $n$-th view feature for the cylinder plane based on $\mathbf{F}_n$. To do this, we project the cylindrical coordinates of all discrete points on the cylinder plane to Cartesian coordinates, which are further projected to 2D camera pixel coordinates using the coordinate of the central point $\mathbf{c}_u^x$, the extrinsic parameter $\mathbf{W}_n^e$, and the intrinsic parameter $\mathbf{P}_n$ of the $n$-th camera. The projected points outside the feature $\mathbf{F}_n$ are abandoned, and only the ones within the feature $\mathbf{F}_n$ are kept. The $n$-th view cylinder plane feature is obtained by bilinear interpolation of $\mathbf{F}_n$ for those kept points. Finally, the cylinder plane feature is produced by overlaying these $N$ views of the cylinder plane feature one-by-one.

Taking the fact into consideration that there exists some overlap between adjacent views, we overlay these $N$ views of the cylinder plane feature in two ways: (1) Overlaying in a clockwise manner. These $N$ views are overlaid from the 0-th view to the $(N-1)$-th view, and the final cylinder plane feature is denoted as $\mathbf{F}_{cy}^+ \in \mathbb{R}^{D_{feat} \times H_u \times W_u}$; (2) Overlaying in a counter-clockwise manner. These $N$ views are overlaid from the $(N-1)$-th view to the 0-th view, and the final cylinder plane feature is denoted as $\mathbf{F}_{cy}^- \in \mathbb{R}^{D_{feat} \times H_u \times W_u}$. During the overlaying procedure, the features for the overlapping points are produced by overwriting the existing ones with the incoming ones. Please refer to Appendix A.6 for the illustration of these two overlay ways. By augmenting the image features with depth and confidence maps generated by the depth estimator (*i.e.*, Metric3D-v2(Hu et al., 2024)), and then processing them through the identical pipeline mentioned above, we generate the cylinder plane feature with depth information, denoted as $\bar{\mathbf{F}}_{cy}^+, \bar{\mathbf{F}}_{cy}^- \in \mathbb{R}^{(D_{feat}+2) \times H_u \times W_u}$.

Note that the parameters $\Delta h$ and $\rho$ are adjustable and are set to different values for occupancy-aware, volume-aware and pixel-aware modules, as well as different camera configurations, which means that the cylinder plane features are slightly different for different modules and different camera configureations. Please refer to Appendix A.12.1 for more details about the settings of $\Delta h$ and $\rho$.

## 3.2 FOREGROUND RECONSTRUCTION

Based on the cylinder plane feature, we can reconstruct the driving scene. However, reconstructing the scene from dense volume is memory- and computation-intensive, as most voxels are empty and can be pruned. To handle this, we reconstruct the foreground and background separately and then fuse them. In addition, the foreground is sparsely reconstructed with the dedicated occupancy-aware, volume-aware, and pixel-aware modules.

The **Occupancy-Aware Module** is designed to distinguish the occupied and empty spaces based on the 2D image features, which is inspired by the work in Wei et al. (2023). The occupancy-aware module is mainly implemented with a Y-shaped Network (termed as YNet$_{occ}$), as shown in Fig. 4. YNet$_{occ}$ takes $\mathbf{F}_{cy}^+$ and $\mathbf{F}_{cy}^-$ as input with a weight-shared dual-branch encoder. The output features of these two branches are fused to be further processed by a single branch decoder. Since most of the features in $\mathbf{F}_{cy}^+$ and $\mathbf{F}_{cy}^-$ are the same, there exists information redundancy between them. To handle this, $\mathbf{F}_{cy}^-$ is flipped before being fed into the network, and the output of the encoder is

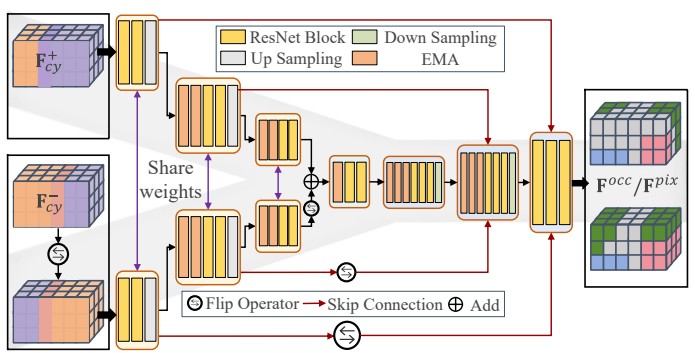
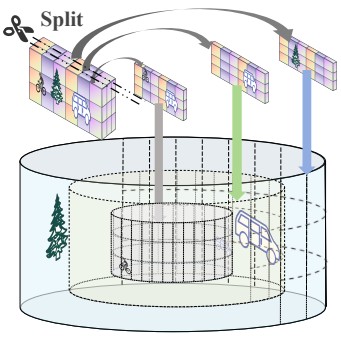

Figure 4: **Architecture of Y-shaped network** for the occupancy-aware module YNet$_{\mathrm{occ}}$ and pixel-aware module YNet$_{\mathrm{pix}}$. The network is mainly implemented based on the ResNet Block (He et al., 2016) and EMA (Ouyang et al., 2023).

Figure 5: The cylinder plane feature group is constructed by splitting the feature in the channel dimension.

flipped back before fusion. Let $\mathbf{F}^{occ} \in \mathbb{R}^{KD_{occ} \times H_u \times W_u}$ be the output feature of YNet$_{\mathrm{occ}}$, whose channel dimension can be evenly divided by $K$. A Cylinder Plane Feature Group (CPFG) $\mathbf{F}^{occ}_{cy} \in \mathbb{R}^{D_{occ} \times K \times H_u \times W_u}$ is obtained by reshaping $\mathbf{F}^{occ}$, as illustrated in Fig. 5. The spatial awareness of the well-pretrained feature extractor is lifted with the help of CPFG.

To judge whether the voxels in the occupancy grid $\mathbf{O}$ are occupied or not, the Cartesian coordinates of all points in $\mathbf{X}_o$ are projected into the cylindrical coordinates (please refer to Appendix A.7 for the details of this projection). Then the occupancy feature $\mathbf{F}^{occ}_o \in \mathbb{R}^{D_{occ} \times L_o \times H_o \times W_o}$ for the occupancy grid $\mathbf{O}$ is obtained by trilinear interpolation of $\mathbf{F}^{occ}_{cy}$. Based on the occupancy feature $\mathbf{F}^{occ}_o$, a 3D occupancy probability map $\hat{\mathbf{P}}_{3D} \in \mathbb{R}^{2 \times L_o \times H_o \times W_o}$ and a 2D BEV occupancy probability map $\hat{\mathbf{P}}_{2D} \in \mathbb{R}^{H_o \times W_o}$ are produced by two lightweight MLP-based heads. In the inference stage, the indices of occupied voxels are determined by $\mathcal{I}_v = \{\mathbf{i} = (l_v, h_v, w_v) | \hat{\mathbf{P}}_{3D}[0, \mathbf{i}] < \hat{\mathbf{P}}_{3D}[1, \mathbf{i}]\}$.

The **Volume-Aware Module** is used to generate 3D Gaussians for the occupied voxels, which renders the outline of the foreground. As shown in Fig. 6, it is implemented with an X-shaped Network (termed as XNet$_{\mathrm{vol}}$). Similar to the YNet$_{\mathrm{occ}}$, XNet$_{\mathrm{vol}}$ takes the cylinder plane feature $\mathbf{F}^+_{cy}$ and $\mathbf{F}^-_{cy}$ as input with a weight-shared dual-branch encoder. But differently, the fused feature of the dual-branch is fed into a dual-branch decoder, where the two branches produce the geometry feature $\mathbf{F}^{geo} \in \mathbb{R}^{KD_{geo} \times H_u \times W_u}$ and the appearance feature $\mathbf{F}^{app} \in \mathbb{R}^{KD_{app} \times H_o \times W_o}$ independently. The two features $\mathbf{F}^{geo}$ and $\mathbf{F}^{app}$ are further reshaped into the geometry CPFG $\mathbf{F}^{geo}_{cy} \in \mathbb{R}^{D_{geo} \times K \times H_u \times W_u}$ and the appearance CPFG $\mathbf{F}^{app}_{cy} \in \mathbb{R}^{D_{app} \times K \times W_o}$. Similar to the occupancy feature $\mathbf{F}^{occ}_o$, the geometry feature $\mathbf{F}^{geo}_o \in \mathbb{R}^{D_{geo} \times L_o \times H_o \times W_o}$ and appearance feature $\mathbf{F}^{app}_o \in \mathbb{R}^{D_{app} \times L_o \times H_o \times W_o}$ for the occupancy grid are obtained by trilinear interpolation of $\mathbf{F}^{geo}_{cy}$ and $\mathbf{F}^{app}_{cy}$, respectively.

For an occupied voxel with the index $\mathbf{i} \in \mathcal{I}_v$, $G_v$ 3D Gaussians $\mathcal{G}_{\mathbf{i}} = \{\mathbf{X}_o[:, \mathbf{i}] + \Delta\mathbf{x}^g_{\mathbf{i}}, \mathbf{c}^g_{\mathbf{i}}, \mathbf{\Sigma}^g_{\mathbf{i}}, \alpha^g_{\mathbf{i}}\}^{G_v-1}_{g=0}$ are generated based on the geometry and appearance features, where $\mathbf{X}_o[:, \mathbf{i}] + \Delta\mathbf{x}^g_{\mathbf{i}}, \mathbf{c}^g_{\mathbf{i}}, \mathbf{\Sigma}^g_{\mathbf{i}}$ and $\alpha^g_{\mathbf{i}}$ are the spatial position, spherical harmonics coefficients, anisotropic covariance matrix, and opacity of the $g$-th 3D Gaussian, respectively. Specifically, the spatial offset $\{\Delta\mathbf{x}^g_{\mathbf{i}}\}^{G_v-1}_{g=0}$ is obtained by a lightweight MLP based on $\mathbf{F}^{geo}_o[:, \mathbf{i}]$, while the rest Gaussian parameters $\{\mathbf{c}^g_{\mathbf{i}}, \mathbf{\Sigma}^g_{\mathbf{i}}, \alpha^g_{\mathbf{i}}\}^{G_v-1}_{g=0}$ are obtained by another lightweight MLP based on $\mathbf{F}^{app}_o[:, \mathbf{i}]$. We denote 3D Gaussians for all occupied voxels as $\mathcal{G}_{\mathcal{I}_v} = \cup_{\mathbf{i} \in \mathcal{I}_v} \mathcal{G}_{\mathbf{i}}$.

The **Pixel-Aware Module** is used to generate additional 3D Gaussians for the texture refinement of the scene. Similar to the occupancy-aware module, the pixel-aware module is also implemented with a Y-shaped network (termed as YNet$_{\mathrm{pix}}$). Different from YNet$_{\mathrm{occ}}$, YNet$_{\mathrm{pix}}$ takes $\bar{\mathbf{F}}^+_{cy}$ and $\bar{\mathbf{F}}^-_{cy}$ as input and utilizes the original attention instead of EMA. Since YNet$_{\mathrm{pix}}$ is designed for the texture refinement, the input/output of YNet$_{\mathrm{pix}}$ is upsampled with the factor $k_i/k_o$ before/after being processed by YNet$_{\mathrm{pix}}$ to make the produced feature have higher spatial resolution ($k_i \times k_o = 4$).

Let $\mathbf{F}^{pix} \in \mathbb{R}^{D_{pix} \times 4H_u \times 4W_u}$ be the upsampled output of YNet$_{\mathrm{pix}}$, and $\mathcal{I}_p = \{\mathbf{i} = (h_u, w_u) | 0 \leq h_u < 4H_u, 0 \leq w_u < 4W_u, \}$ be the indices of all pixels in $\mathbf{F}^{pix}$, we generate $G_p$ 3D Gaussians

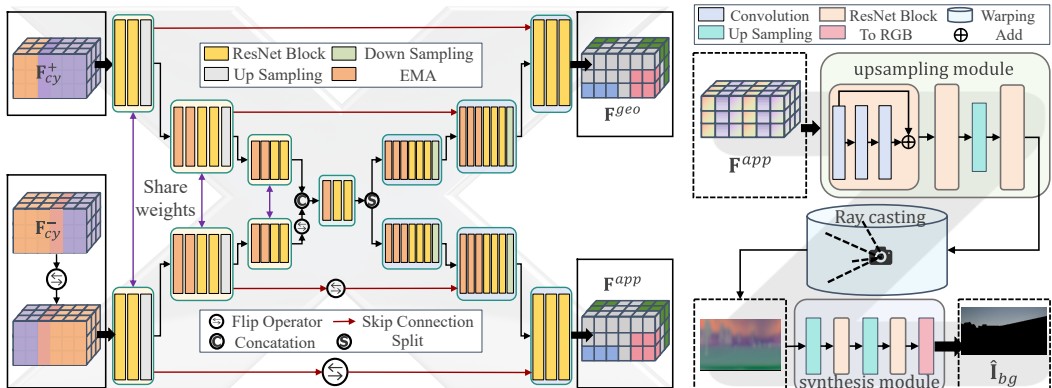

Figure 6: **Architecture of XNet**$_{\text{vol}}$, which is composed of a dual-branch encoder for downsampling and a dual-branch decoder for appearance and geometric feature upsampling.

Figure 7: **Overview of the Z-Net architecture.** It includes upsampling, ray casting and synthesis.

$\mathcal{G}_{\mathbf{i}} = \{(\mathbf{x}_{\mathbf{i}} + \Delta\mathbf{x}_{\mathbf{i}}^g, \mathbf{c}_{\mathbf{i}}^g, \boldsymbol{\Sigma}_{\mathbf{i}}^g, \alpha_{\mathbf{i}}^g)\}_{g=0}^{G_p-1}$ for each pixel $\mathbf{i} \in \mathcal{I}_p$, where $\mathbf{x}_{\mathbf{i}}$ is the projected Cartesian coordinates of pixel $\mathbf{i}$ with the help of the estimated depth from the depth estimator. The Gaussian parameters $\{\Delta\mathbf{x}_{\mathbf{i}}^g, \mathbf{c}_{\mathbf{i}}^g, \boldsymbol{\Sigma}_{\mathbf{i}}^g, \alpha_{\mathbf{i}}^g\}_{g=0}^{G_p-1}$ are estimated by a lightweight MLP based on $\mathbf{F}^{pix}[:, \mathbf{i}]$. The 3D Gaussians for all pixels are denoted as $\mathcal{G}_{\mathcal{I}_p} = \cup_{\mathbf{i} \in \mathcal{I}_p} \mathcal{G}_{\mathbf{i}}$.

The foreground is represented by 3D Gaussians as $\mathcal{G}_{fg} = \mathcal{G}_{\mathcal{I}_v} \cup \mathcal{G}_{\mathcal{I}_p}$. Given a target view with the camera parameters $\mathbf{W}_t^e$ and $\mathbf{P}_t$, and spatial resolution $H_t \times W_t$, we can obtain the rendered foreground image $\hat{\mathbf{I}}_{fg} \in \mathbb{R}^{3 \times H_t \times W_t}$, $\alpha$-map $\hat{\mathbf{A}}_{fg} \in \mathbb{R}^{H_t \times W_t}$ and depth map $\hat{\mathbf{D}}_{fg} \in \mathbb{R}^{H_t \times W_t}$ with the rasterization rendering of $\mathcal{G}_{fg}$.

## 3.3 BACKGROUND RECONSTRUCTION AND FUSION

The background is generated in the 2D space with a Z-shaped Network (termed as ZNet$_{\text{bg}}$) and fused with the rendered foreground image $\hat{\mathbf{I}}_{fg}$. As shown in Fig. 7, ZNet$_{\text{bg}}$ takes the appearance feature $\mathbf{F}^{app}$ produced by XNet$_{\text{vol}}$ as input, and produces the generated background image. Similar to the pixel-aware module YNet$_{\text{pix}}$, ZNet$_{\text{bg}}$ also needs to generate the detailed texture of the background (*i.e.*, the sky and clouds). Taking this into consideration, the background image is generated at a higher resolution (in detail, $4H_t \times 4W_t$) and downsampled to the target resolution (*i.e.*, $H_t \times W_t$) and fused with the foreground image.

The generation of the background is divided into three steps within ZNet$_{\text{bg}}$. (1) Upsample the input feature by $2\times$ with a ResNet-based (He et al., 2016) upsampling module. (2) Cast rays from the camera center of the target view through each pixel of the target resolution $H_t \times W_t$. The intersection points of these rays and the upsampled feature in step (1) are computed (refer to Appendix A.7), which are used to bilinearly sample the target feature (with the resolution of $H_t \times W_t$) for the target view. (3) Generate the background image $\hat{\mathbf{I}}_{bg} \in \mathbb{R}^{3 \times 4H_t \times 4W_t}$ using a StyleGAN-based (Karras et al., 2021) synthesis module based on the target feature sampled in step (2).

To fuse the background image with the foreground image, the background image $\hat{\mathbf{I}}_{bg}$ is downsampled to the target resolution $\hat{\mathbf{I}}'_{bg} \in \mathbb{R}^{3 \times H_t \times W_t}$. Then, $\alpha$-blending is adopted to fuse the downsampled background image with the foreground image $\hat{\mathbf{I}} = \hat{\mathbf{I}_{fg}} + (1 - \hat{\mathbf{A}}_{fg}) \otimes \hat{\mathbf{I}}'_{bg}$.

## 3.4 OPTIMIZATION OF XYZCYLINDER

The optimization of XYZCylinder is divided into two stages. (1) Training the occupancy-aware module by supervising the 3D and 2D occupancy probability maps $\hat{\mathbf{P}}_{3D}$ and $\hat{\mathbf{P}}_{2D}$. The cross-entropy loss, semantic loss (Cao & De Charette, 2022), and geometric loss (Cao & De Charette, 2022) are used for $\hat{\mathbf{P}}_{3D}$, while the BEV loss (Hou et al., 2024) is adopted for $\hat{\mathbf{P}}_{2D}$. The ground-truth occupancy map is generated by aggregating semantic point clouds using Poisson reconstruction, followed by a voxelization operation. (2) Training the rest modules with the frozen occupancy-

aware module by supervising the rendered image $\hat{\mathbf{I}}$, $\alpha$-map $\hat{\mathbf{A}}_{fg}$, and depth $\hat{\mathbf{D}}_{fg}$. Specifically, $\hat{\mathbf{I}}$ is supervised by the reconstruction L1 loss and perceptual similarity loss (Zhang et al., 2018). $\hat{\mathbf{A}}_{fg}$ is only supervised by the reconstruction L1 loss, and the ground-truth of the $\alpha$-map is obtained by LISA (Lai et al., 2024). $\hat{\mathbf{D}}_{fg}$ is supervised by the reconstruction L1 loss and Pearson depth loss (Xiong et al., 2023), and the ground-truth depth map is obtained by the depth estimator. Please refer to Appendix A.8 for more details of data preprocessing and Appendix A.12.1 for detailed losses.

# 4 EXPERIMENTS

**Baselines.** We select recent works, including SplatterImage (Szymanowicz et al., 2024), PixelSplat (Charatan et al., 2024), MVSplat (Chen et al., 2024), DepthSplat (Xu et al., 2025), DrivingForward (Tian et al., 2025), Omni-Scene (Wei et al., 2025), and 6Img-to-3D (Gieruc et al., 2024) as the baselines for ego-forward and ego-inward tasks. Implementation details and modifications for the baselines on both datasets are provided in Appendix A.9.2.

**Datasets.** We follow Omni-Scene (Wei et al., 2025) and 6Img-to-3D (Gieruc et al., 2024) to evaluate ego-forward and ego-inward reconstruction on nuScenes and Carla-Centric, respectively. In both evaluation settings, 6 views are used without specification. In addition, Waymo (Sun et al., 2020), Pandaset (Xiao et al., 2021), ONCE (Mao et al., 2021), and Argoverse (Chang et al., 2019) are used for zero-shot evaluation. For more details, please refer to Appendix A.8

**Metrics.** We evaluate our method's photometric quality using the standard metrics of PSNR, SSIM, and LPIPS. To assess geometric accuracy, we compute the PCC between the rendered depth maps and predicted depth maps. The detailed formulas for these metrics can be found in Appendix A.9.1.

**Comparison with Baselines.** Table 1 presents the quantitative comparison between the proposed XYZCylinder and several baselines. Overall, XYZCylinder achieves all the best metrics on both datasets. Beyond its superior hybrid representation and advanced architecture, the superior performance of our method also stems from two aspects. First, by decoupling the reconstruction of foreground and background, we prevent the foreground 3D Gaussians $\mathcal{G}_{fg}$ from being misplaced into the background region (*i.e.*, sky). Second, the frozen occupancy-aware module can constrain the optimization space of the volume-aware module, benefiting the generation of 3D Gaussians $\mathcal{G}_{\mathcal{I}_v}$ which are vital to the geometric accuracy.

Interestingly, all methods achieve substantially better performance on nuScenes than on Carla-Centric. The reason is that the viewpoint shifts between the training and test sets of nuScenes are smaller than those of Carla-Centric, reducing the demand for geometric reconstruction capability. Taking the PSNR on nuScenes for example, some pixel-based methods (*e.g.*, DepthSplat and DrivingForward) achieve better PSNR than the volume-based method (*i.e.*, 6Img-to-3D), and achieve considerable PSNR with

Table 1: **Quantitative comparison of our model against the baselines.** The best, second-best, and third-best results are marked with colors.

| Models | nuScenes (ego-forward) | | | | Carla-Centric (ego-inward) | | | |
|---|---|---|---|---|---|---|---|---|
| | PSNR↑ | LPIPS↓ | SSIM↑ | PCC↑ | PSNR↑ | LPIPS↓ | SSIM↑ | PCC↑ |
| SplatterImage | 17.31 | 0.661 | 0.442 | 0.027 | 13.04 | 0.708 | 0.448 | 0.180 |
| PixelSplat | 21.33 | 0.376 | 0.607 | 0.077 | 14.67 | 0.565 | 0.412 | 0.554 |
| MVSplat | 21.87 | 0.342 | 0.621 | 0.201 | 15.32 | 0.507 | 0.457 | 0.566 |
| DepthSplat | 23.19 | 0.339 | 0.675 | 0.431 | 16.27 | 0.503 | 0.508 | 0.581 |
| DrivingForward | 23.84 | 0.280 | 0.739 | 0.437 | 15.38 | 0.541 | 0.445 | 0.696 |
| 6Img-to-3D | 20.74 | 0.650 | 0.560 | 0.570 | 17.33 | 0.485 | 0.620 | 0.765 |
| Omni-Scene | 24.11 | 0.242 | 0.734 | 0.816 | 15.54 | 0.558 | 0.462 | 0.551 |
| XYZCylinder | 24.97 | 0.231 | 0.750 | 0.887 | 18.40 | 0.359 | 0.622 | 0.817 |

volume-pixel-based methods (*i.e.*, Omni-Scene and XYZCylinder). Nevertheless, the geometric accuracy of pixel-based methods, particularly in depth estimation (*i.e.*, PCC), remains limited. Although both our XYZCylinder and Omni-Scene are volume-pixel-based methods, Omni-Scene suffers from suboptimal performance. This is primarily because its volumetric branch relies on feature injection from the pixel branch and employs volumetric sampling lacking geometric constraints.

The qualitative comparisons of different methods further highlight the superiority of XYZCylinder. As shown in Fig. 8 and Fig. 9, XYZCylinder produces significantly better visual quality on Carla-Centric with finer texture details and more accurate geometry, while other methods are plagued by the artificial holes (*i.e.*, white regions). On nuScenes, XYZCylinder further demonstrates its advantages through enhanced geometric completeness (*e.g.*, reconstructing a complete traffic light), superior texture fidelity with fewer artifacts and sharper text, and higher geometric accuracy, free from ghosting or distortion. Extended analyses and further discussions are available in Appendix A.5.

**Ablation on the Importance of Main Designs.** Here, we show the importance of the separate re-

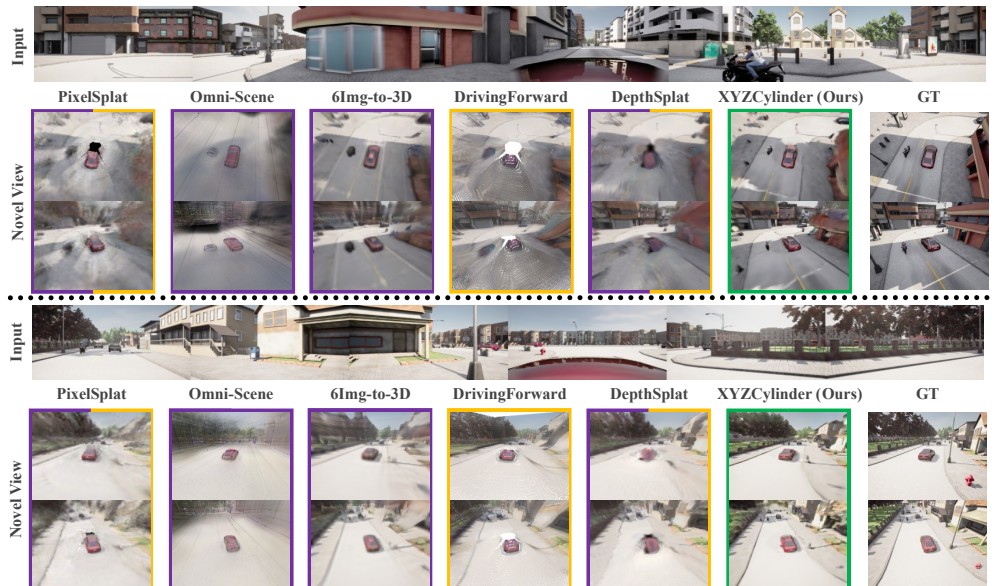

**Figure 8: Qualitative comparison of different methods on Carla-Centric.** Yellow boxes indicate scene void phenomena (*i.e.*, white regions), purple boxes denote blurring artifacts, and green boxes signify that our XYZCylinder successfully circumvents both aforementioned issues.

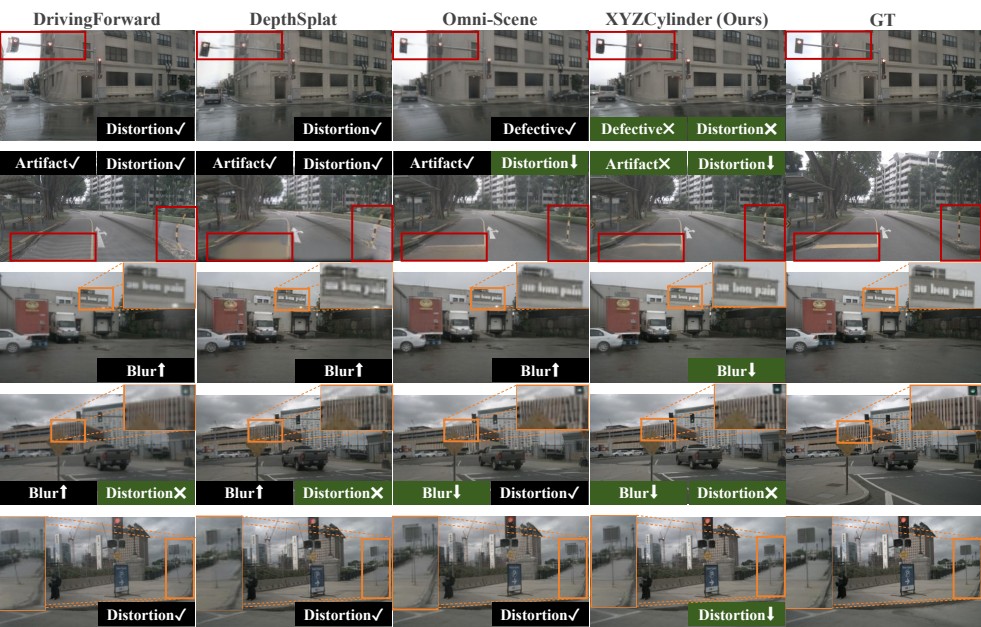

**Figure 9: Qualitative comparison of different methods on nuScenes.** The baseline method exhibits significant geometric distortions (warping artifacts), structural defects (incomplete geometries), texture blurring, and visual artifacts. In contrast, our XYZCylinder produces superior visual quality with much reduced artifacts.

construction of foreground and background, as well as the necessity of the occupancy/volume/pixel-aware module for foreground reconstruction. Results are shown in Table 2. When the background (*i.e.*, sky and clouds) is treated as the foreground and reconstructed by 3D Gaussians (w/o. Separate Reconstruction), or the volume-aware

Table 2: **Ablation study on our main designs.** The best, second-best, and third-best results are marked.

| Models | nuScenes (ego-forward) | | | | Carla-Centric (ego-inward) | | | |
|---|---|---|---|---|---|---|---|---|
| | PSNR | LPIPS | SSIM | PCC | PSNR | LPIPS | SSIM | PCC |
| w/o. Separate | 23.251 | 0.280 | 0.725 | 0.854 | 17.58 | 0.371 | 0.611 | 0.797 |
| w/o. Occupancy | Out of Memory | | | | | | | |
| w/o. Volume | 23.92 | 0.256 | 0.722 | 0.886 | 15.67 | 0.589 | 0.455 | 0.694 |
| w/o. Pixel | 22.41 | 0.338 | 0.661 | 0.667 | 18.19 | 0.367 | 0.616 | 0.521 |
| XYZCylinder | 24.97 | 0.231 | 0.750 | 0.887 | 18.40 | 0.359 | 0.622 | 0.817 |

and pixel-aware modules are removed (w/o. Volume and w/o. Pixel), all metrics on two datasets are reduced, indicating the necessity of the key designs. In addition, XYZCylinder runs out of memory if the occupancy-aware module is removed by treating all voxels as occupied (w/o. Occupancy), which means that the occupancy-aware module can save memory efficiently. The full model's superior performance validates our hybrid representation's effect on reconstruction accuracy.

**Ablation on the Design of X-shape and Y-shape Networks.** The common designs in both networks are the dual-branch encoder and the flip operation of the counter-clockwise cylinder plane feature. To show the effectiveness of such designs, we remove the flip operation (w/o. X-flip and w/o. Y-flip), and replace the dual-branch encoder with a single-branch encoder which takes the clockwise cylinder plane feature as input (w/o. X-dual enc and w/o. Y-dual enc).

In addition, the dual-branch decoder in the X-shape network ($XNet_{vol}$) disentangles the geometry and appearance information of the scene, which is beneficial for the background reconstruction. The effectiveness of such a design is validated by replacing the dual-branch decoder with a single-branch decoder (w/o. X-dual dec), where the single output feature is used for the generation of all parameters for 3D Gaussians in $\mathcal{G}_{\mathcal{T}_v}$ and the generation of background. Results are shown in Table 3. As we

Table 3: **Ablation study on the design of X-shape and Y-shape network.** The best, second-best, and third-best results are marked with colors.

| Models | nuScenes(ego-forward) | | | | Carla-Centric(ego-inward) | | | |
|---|---|---|---|---|---|---|---|---|
| | PSNR | LPIPS | SSIM | PCC | PSNR | LPIPS | SSIM | PCC |
| w/o. X-flip | 24.88 | 0.240 | 0.747 | 0.890 | 18.28 | 0.381 | 0.603 | 0.802 |
| w/o. Y-flip | 24.33 | 0.245 | 0.734 | 0.881 | 18.36 | 0.364 | 0.620 | 0.793 |
| w/o. X-dual enc | 24.16 | 0.260 | 0.721 | 0.876 | 18.21 | 0.377 | 0.607 | 0.793 |
| w/o. Y-dual enc | 24.04 | 0.256 | 0.726 | 0.887 | 18.34 | 0.360 | 0.618 | 0.796 |
| w/o. X-dual dec | 24.31 | 0.239 | 0.741 | 0.885 | 18.33 | 0.367 | 0.620 | 0.807 |
| XYZCylinder | 24.97 | 0.231 | 0.750 | 0.887 | 18.40 | 0.359 | 0.622 | 0.817 |

can see, all the counterpart models achieve inferior performance to XYZCylinder, demonstrating the effectiveness of our designs. In addition, the models w/o. X-dual enc and w/o. Y-dual enc produce much worse results than others, indicating that the construction of the CPFG relies on better feature extraction.

**Zero-shot on Various Driving Datasets.** The baseline methods lack the generalization capabilities across different datasets, due to the dedicated feature representations for a given number of views. Differently, the proposed XYZCylinder is robust to the number of views thanks to the design of UCCM, which only needs to adjust the construction parameters ($\rho$ and $\Delta h$) of the Cylinder Plane. Fig. 10 presents the reconstructed driving scenes on Waymo (Sun et al., 2020), Pandaset (Xiao et al., 2021), ONCE (Mao et al., 2021), and Argoverse (Chang et al., 2019) using the nuScenes-trained model. Please refer to Appendix A.11 for more zero-shot results.

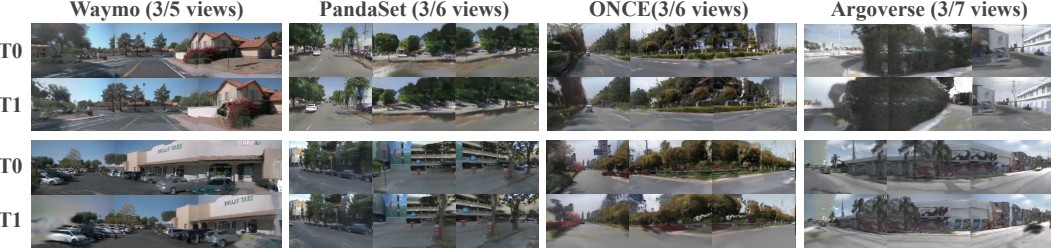

Figure 10: **Zero-shot results on other datasets.** The XYZCylinder, trained solely on nuScenes, is evaluated. Only the reconstructed 3 views are presented due to the limited space.

## 5 CONCLUSION

We present XYZCylinder, a feedforward framework designed for sparse 3D reconstruction of driving scenes. XYZCylinder employs a novel Unified Cylinder Lifting Method to enhance generalization capability and reconstruction accuracy. Our model utilizes an explicit, deterministic view transformation and a hybrid representation to reconstruct foreground and background, ensuring both high-fidelity 3D reconstruction and camera parameter compatibility. Extensive experiments on nuScenes and Carla-Centric as well as the zero-shot results demonstrate the outstanding reconstruction and generalization capabilities of XYZCylinder, revealing its enormous potential for both autonomous driving perception enhancement (with ego-forward cameras) and collaborative perception enhancement (with ego-inward cameras) within a unified architecture. The success of our model further expands the capability boundary of the feedforward models in generating complete 3D scenes and constructing simulation environments for 3D autonomous driving systems.

ETHICS STATEMENT

Apart from the publicly available datasets for autonomous driving, our work conducts a new dataset entirely through simulation. All tools and resources used for data generation, including the simulator and its assets, are publicly available. By leveraging a synthetic data approach, we circumvent ethical concerns commonly associated with real-world data collection. Specifically, our methodology does not involve human subjects, collect any personally identifiable information, or handle sensitive data. Therefore, issues of privacy, consent, and data anonymization are not applicable to this work.

REPRODUCIBILITY STATEMENT

We are committed to the reproducibility of our work and will release the full source code, pre-trained models, and experiment scripts publicly upon publication. To facilitate the review process, we provide extensive implementation details in the appendix. The mathematical derivations for key operations in the main text are detailed in Appendix A.6 and Appendix A.7. The construction process and preprocessing steps for all datasets are described in Appendix A.8. A complete account of our model's implementation, including network architecture and other specifics, is provided in Appendix A.9.

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

APPENDIX CONTENTS

# A APPENDIX

## A.1 LLM USAGE

In the preparation of this manuscript, we utilized a large language model (LLM), specifically Gemini-2.5Pro, for the sole purpose of language polishing. The LLM was employed to improve grammar, clarity, and readability. It was not used for generating scientific ideas, conducting data analysis, or drafting the core content of the paper. The authors have carefully reviewed and edited all suggestions made by the LLM and take full responsibility for the final content of this manuscript.

## A.2 PRELIMINARIES

**3D Gaussian Splatting.** Representing 3D Gaussians as ellipsoids establishes an isomorphism between them. Consequently, a collection of 3D Gaussians can model arbitrary 3D geometry. Each Gaussian is parameterized by its spatial position (mean) $\mathbf{X}$, an anisotropic covariance matrix $\mathbf{\Sigma}$, an opacity $\alpha$, and its spherical harmonics coefficients $\mathbf{c}$. The covariance matrix $\mathbf{\Sigma}$ determines the ellipsoid's geometry and is decomposed into a scaling matrix $\mathbf{S}$ and a rotation matrix $\mathbf{\Psi}$. As illustrated in Eq. (1), the final shape is formed by applying an axis-aligned scaling followed by a rotation.

$$G(\mathbf{X}) = e^{-\frac{1}{2}(\mathbf{X})^T \mathbf{\Sigma}^{-1}(\mathbf{X})}, \quad where \quad \mathbf{\Sigma} = \mathbf{\Psi} \mathbf{S} \mathbf{S}^T \mathbf{\Psi}^T \tag{1}$$

The ellipsoids are rendered onto 2D images using a fast rasterization pipeline that projects each ellipsoid onto the image plane. The 2D covariance for each projection is computed using the viewing transformation $\mathbf{W}$ and the Jacobian $\mathbf{J}$ of the projective transformation, as detailed in Eq. (2).

$$\bar{\mathbf{\Sigma}} = \mathbf{J} \mathbf{W} \mathbf{\Sigma} \mathbf{W}^T \mathbf{J}^T \tag{2}$$

The final pixel color is synthesized via the alpha compositing technique from Eq. (3). This involves blending the contributions of $N$ Gaussians that overlap the pixel. These Gaussians are first sorted by depth, and for the $\mathbf{i}$-th Gaussian, its color $\mathbf{c_i}$ is evaluated from its Spherical Harmonics (SH) coefficients according to the viewing direction. The colors are then blended in front-to-back order to yield the final pixel value.

$$C = \sum_{\mathbf{i} \in N} \mathbf{c_i} \alpha_{\mathbf{i}} \prod_{\mathbf{j}=1}^{\mathbf{i}-1} (1 - \alpha_{\mathbf{j}}), \tag{3}$$

## A.3 DETAILED RELATED WORK

**Feedforward Reconstruction Models.** Feedforward reconstruction methods can be broadly categorized based on their underlying 3D scene representation. The first category comprises pixel-based models (Wang et al., 2024a; Tang et al., 2025; Wang et al., 2025a; Charatan et al., 2024; Chen et al., 2024; Xu et al., 2025; Zhang et al., 2025; Fei et al., 2024a; Miao et al., 2025; Liu et al., 2024b; Zheng et al., 2025; Fei et al., 2024b; Wang et al., 2024b; 2025b; Liu et al., 2025b; Xiao et al., 2025; Li et al., 2024b; Min et al., 2024; Smart et al., 2024), which typically reconstruct scenes as explicit point clouds or, more recently, Gaussian splats. These approaches operate by predicting per-pixel attributes like depth and local offsets, enabling them to generate dense and detailed geometry, particularly in forward-facing scenarios with substantial view overlap. However, their reliance on direct 2D-to-3D projection makes them vulnerable in challenging outward-facing settings, such as autonomous driving. In these cases, they often produce reconstructions characterized by voids in unobserved regions and struggle to reason about complex occlusions. In contrast, volumetric methods (Gieruc et al., 2024; Ren et al., 2024; Zou et al., 2024; Liu et al., 2025a) represent the scene implicitly within a continuous or discrete volume (e.g., a neural radiance field or a voxel grid). This inherent volumetric nature ensures the generation of topologically complete and hole-free geometry, offering a natural mechanism for handling occlusions. The trade-off, however, is that these representations often yield overly smooth surfaces, struggling to capture the high-frequency textural and geometric details at which pixel-based methods excel. To harness the complementary strengths of both paradigms, hybrid methods like Omni-Scene (Wei et al., 2025) have emerged. These approaches aim to integrate the detail-rich output of pixel-based techniques with the completeness of volumetric representations. Our work aligns with this hybrid philosophy but introduces a key architectural divergence. Whereas Omni-Scene employs a pixel-guided strategy in which one representation heavily influences the other, our model utilizes independent, parallel prediction branches

for each representation. This decoupled design allows each branch to specialize, culminating in a final reconstruction that is more geometrically complete and more finely detailed than previous methods.

**2D Feature Lifting.** The task of lifting 2D image features into a 3D spatial representation is a cornerstone of modern 3D perception. Pioneering works in this area often rely on explicit depth estimation. For instance, CaDDN (Reading et al., 2021) generates a categorical depth distribution for each pixel to cast 2D features into a 3D volume. Similarly, LSS (Philion & Fidler, 2020) generates a BEV map by splatting per-pixel features into a grid, guided by a predicted continuous depth distribution. More recent paradigms have shifted towards attention mechanisms to establish this 2D-to-3D correspondence. BEVFormer (Li et al., 2024c) pioneered the use of spatial cross-attention to query and aggregate multi-camera 2D features into a unified BEV representation. TPVFormer (Huang et al., 2023) extends this concept by constructing a more comprehensive Tri-Perspective View (TPV) representation, populating three orthogonal planes from the 2D inputs. Further advancing this line of work, DFA3D (Li et al., 2023) integrates depth-aware sampling into a 3D deformable attention mechanism for more precise feature lifting. Diverging from these approaches, which learn to lift features via explicit depth prediction or complex attention queries, our method introduces a novel, principled restructuring technique. We leverage the inherent spatial coherence within 2D feature maps, reordering the channel dimensions to directly form a semantically rich pillar-based representation. This pillar field serves as an effective and efficient intermediate representation for downstream 3D tasks.

## A.4 TASK SETTINGS

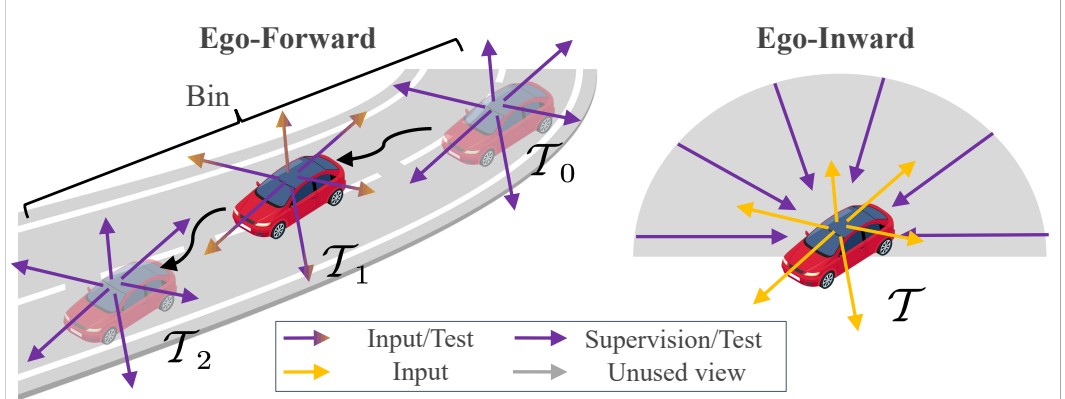

Figure 11: **Two reconstruction settings for comprehensive assessment.** We introduce two tasks for autonomous driving scene reconstruction: the Ego-Forward setting, evaluated on the nuScenes dataset, and the Ego-Inward setting, evaluated on a custom Carla-Centric dataset. These settings are designed to probe different model capabilities and serve distinct downstream applications. The Ego-Forward task targets common driving scenarios, producing assets suitable for testing and simulating perception algorithms. In contrast, the Ego-Inward task generates a comprehensive, omnidirectional scene representation. This holistic view is particularly valuable for applications involving heterogeneous data sources, such as Vehicle-to-Everything (V2X) collaboration and air-to-ground joint perception.

The ego-forward setting utilizes forward-facing sequences from autonomous driving scenarios to assess generalizable reconstruction from sparse views, thereby measuring a model's potential of perception in standard driving situations. In contrast, the ego-inward setting focuses on the scene itself, evaluating performance under large viewpoint variations. This latter test probes a model's robustness, its ability to handle heterogeneous data sources, and its capacity for enhancing perception features across wide-span views.

As illustrated in Fig. 11, we define two distinct task settings. The ego-forward setting evaluates forward-facing sequence reconstruction: the model processes six input views from time $\mathcal{T}_1$ and is supervised using ground-truth views from a temporal window spanning $\mathcal{T}_0$, $\mathcal{T}_1$, and $\mathcal{T}_2$. In contrast, the ego-inward setting assesses spatial generalization. Given the same six views from time $\mathcal{T}$, the

918
919
920
921
922
923
924
925
926
927
928
929
930
931
932
933
934
935
936
937
938
939
940
941
942
943
944
945
946
947
948
949
950
951
952
953
954
955
956
957
958
959
960
961
962
963
964
965
966
967
968
969
970
971

model is evaluated on its ability to synthesize novel views from a surrounding hemisphere at that identical timestep.

The radar chart in Fig. 12 illustrates the strong unification capability of our model, where a larger area signifies superior comprehensive performance. It demonstrates that our model achieves excellent overall results on two reconstruction tasks under different settings.

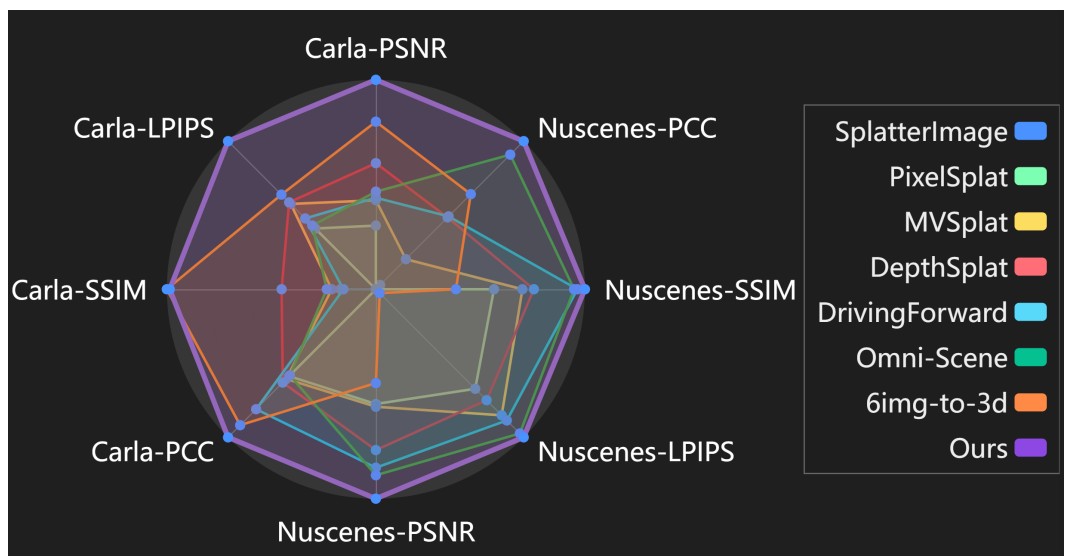

Figure 12: **Comparison of model performance via a radar chart.** The area of the radar chart represents the comprehensive performance of each model. A larger enclosed area indicates a stronger capability to unify the two experimental settings.

## A.5 DISCUSSION

**Qualitative Insights into the Inherent Advantages of Our Approach.** Unlike previous approaches such as Omni-Scene (Wei et al., 2025) and 6Img-to-3D (Gieruc et al., 2024), we introduce a more interpretable and parameter-free Unified Cylinder Camera Model (UCCM) for handling camera transformations. This design eliminates the need for complex attention mechanisms to learn large angular shifts between views. The network architecture operates on the Cylinder Plane Feature, making it inherently compatible with established 2D feature enhancement techniques, such as the patching strategy from Masked Autoencoders (He et al., 2022). Furthermore, visualizations of the Cylinder Plane Feature reveal that while projection artifacts do create black borders, these regions are proportionally small. This characteristic allows us to frame the task as a feature inpainting problem, drawing a direct parallel to the MAE framework. In essence, our model functions as an autoencoder. During supervised training, it learns to progressively fill these border regions with meaningful semantic content, a process that simultaneously strengthens the representation of the learned features.

**Application.** Fundamentally, our model is designed as a direct 2D-to-3D lifting framework. This architectural choice provides a straightforward yet powerful pathway to elevate existing 2D autonomous driving scene generation methods into the 3D domain. For instance, state-of-the-art 2D generation models like MagicDrive-v2 (Gao et al., 2024b), which excel at producing diverse and realistic driving scenarios, could be seamlessly integrated with our approach. By leveraging their powerful 2D backbones and applying our lifting module, we can directly generate high-fidelity, geometrically consistent 3D scenes without the need to retrain a large-scale 3D generator from scratch. This not only democratizes 3D scene generation but also significantly accelerates the development cycle. Beyond its application in generation, our work pioneers a new paradigm for 2D-to-3D lifting in the context of autonomous driving. By re-framing the geometric transformation in a more interpretable and efficient manner, we unlock latent potential for various downstream perception tasks. For example, our method could enhance 3D object detection by providing richer geometric cues or improve BEV (Bird's-Eye-View) segmentation by warping 2D features into a more robust and

spatially aware representation. We believe this novel approach opens up promising new avenues for research, offering a flexible and effective bridge between the mature 2D vision ecosystem and the burgeoning field of 3D autonomous driving perception.

**Future Work.** Our proposed framework opens up several promising avenues for future research. We outline four key directions below:

a) *Extension to 4D Reconstruction for Dynamic Objects*: Our current model primarily focuses on static scenes. A natural and critical next step is to extend our 2D-to-3D lifting paradigm to the temporal domain, enabling 4D reconstruction. This would involve incorporating temporal cues to model the motion and deformation of dynamic agents, such as vehicles and pedestrians.

b) *End-to-End Semantic and Instance Segmentation in 3D*: While our method generates geometrically accurate 3D structures, integrating rich semantic understanding is crucial. We plan to explore end-to-end architectures that jointly perform 3D reconstruction and semantic/instance segmentation. Our interpretable feature-lifting mechanism provides a strong foundation for this, as the lifted 2D semantic features can be directly supervised in the 3D space, potentially leading to more accurate and consistent segmentation in complex urban environments.

c) *Generalizable Material and Texture Reconstruction*: To enhance the realism of generated scenes for simulation and data augmentation, we aim to reconstruct not only geometry but also surface materials and textures. This involves predicting spatially varying Bidirectional Reflectance Distribution Functions (BRDFs) or other appearance models. Our framework's ability to handle complex projective geometry could be adapted to disentangle material properties from illumination effects, paving the way for high-fidelity, relightable 3D asset creation for autonomous systems.

d) *Large-Scale Scene Generation*: This direction involves investigating method for stitching the outputs of multiple, spatially-distinct predictive modules to generate large-scale, cohesive 3D environments. This requires ensuring geometric, semantic, and instance-level consistency across the boundaries of the individual predictions.

## A.6 THE OVERLAYING PROCEDURE

**Definition 1.** *The ordered composition operators* $\gamma(\mathbf{f}_a, \mathbf{f}_b)$ *are defined by the following equations:*

$$\gamma(\mathbf{f}_a, \mathbf{f}_b) = \begin{cases} \mathbf{f}_b, & \mathbf{f}_b \neq 0 \\ \mathbf{f}_a, & \mathbf{f}_b = 0 \end{cases} \tag{4}$$

Fig. 13 provides a schematic illustration of the ordered composition operators applied to autonomous driving data.

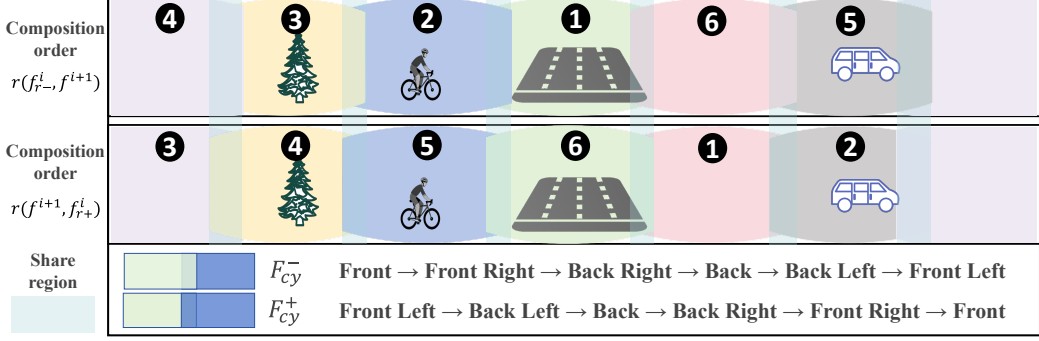

Figure 13: **Effectiveness of directional composition.** By employing clockwise manner and counterclockwise manner, the covered area constitutes only a small fraction of the six images.

A homogeneous point $\mathbf{X}_{cy} = (R_u, \theta, z, 1)^T \in \mathbb{R}^4$ on the cylinder plane is represented in the Cartesian coordinate system as $(x, y, z, 1)^T$. Acquiring image features for the **i**-th discrete points on the cylinder requires projecting them back onto the camera pixel coordinates $\mathbf{X}_{uv}^i \in \mathbb{R}^4$. Using the nor-

malized projected 2D coordinates $\hat{\mathbf{X}}^{\mathbf{i}}_{uv} = (2x^{\mathbf{i}}_{uv}/z^{\mathbf{i}}_{uv} - 1, 2y^{\mathbf{i}}_{uv}/z^{\mathbf{i}}_{uv} - 1) \in \mathbb{R}^2$, we sample the corresponding $\mathbf{i}$-th image feature $\mathbf{F}_{\mathbf{i}} \in \mathbb{R}^{D_{feat}}$ obtained by Radio-v2.5 (Heinrich et al., 2024) via bilinear interpolation to obtain the cylindrical point feature $\mathbf{f}^{\mathbf{i}}_{uv} \in \mathbb{R}^{D_{feat}}$ using $\mathbf{f}^{\mathbf{i}}_{uv} = bilinear(\mathbf{F}_{\mathbf{i}}, \hat{\mathbf{X}}^{\mathbf{i}}_{uv})$.

The features of $\mathbf{X}_{cy}$ are constructed via an ordered composition process which is governed by Eq. (5) where $r(\cdot, \cdot)$ is defined above. For $i = 0, \ldots, N - 2$, we have:

$$\mathbf{f}^{\mathbf{i+1}}_{r-} = \gamma(\mathbf{f}^{\mathbf{i}}_{r-}, \mathbf{f}^{\mathbf{i+1}}), \quad \mathbf{f}^{0}_{r-} = \mathbf{f}^{0}_{uv}, \quad \mathbf{f}^{\mathbf{i+1}}_{r+} = \gamma(\mathbf{f}^{\mathbf{i+1}}, \mathbf{f}^{\mathbf{i}}_{r+}), \quad \mathbf{f}^{0}_{r+} = \mathbf{f}^{0}_{uv} \tag{5}$$

The ordered composition process is executed using two distinct traversal orders: clockwise $(+)$ and counter-clockwise $(-)$, as depicted in Fig. 13. For any given point on the Cylinder Plane, these two traversals independently produce final feature vectors, denoted $\mathbf{f}_{r+} = \mathbf{f}^{N-1}_{r+} \in \mathbb{R}^{D_{feat}}$ and $\mathbf{f}_{r-} = \mathbf{f}^{N-1}_{r-} \in \mathbb{R}^{D_{feat}}$, respectively. Applying this process across the entire plane generates two complete feature maps: $\mathbf{F}^{+}_{cy} \in \mathbb{R}^{D_{feat} \times H_u \times W_u}$ and $\mathbf{F}^{-}_{cy} \in \mathbb{R}^{D_{feat} \times H_u \times W_u}$.

## A.7 DERIVATION

**Sampling in Cylinder Plane Feature Group.** For any point $\mathbf{X}_o[;k] = (r_k, \theta_k, z_k)$ within CPFG space with a maximum radius of $R_{max}$ and a minimum radius of $R_{min}$, the geometric relationship is illustrated in Fig. 14 and Fig. 15. Within the annular cylindrical region bounded by a near radius $R_{min}$ and a far radius $R_{max}$, the normalized ratios for the radial component $t_k$ and the normalized angular component $s_k$ are computed as follows:

$$t_k = \frac{r_k - R_{min}}{R_{max} - R_{min}}, \quad s_k = \frac{\pi - \theta_k}{2\pi} \tag{6}$$

The normalization ratio for the height component $p_k$ is linearly dependent on the radial distance $r_k$. This is because the volume is a frustum, where the floor and ceiling heights change linearly with the radius. Based on the principle of similar triangles, we can derive $p_k$ as follows:

$$\frac{r_k}{R_u} = \frac{Z_{cur}}{Z_u}, \quad \frac{(\frac{Z_{cur}}{2} + z_k)}{Z_{cur}} = 1 - p_k \tag{7}$$

Substituting and rearranging the terms, we obtain:

$$p_k = 1 - \frac{(\frac{Z_{cur}}{2} + z_k)}{Z_{cur}} = 1 - (\frac{1}{2} + \frac{z_k}{Z_{cur}}) = \frac{1}{2} - \frac{z_k}{Z_{cur}} = \frac{1}{2} - \frac{z_k R_u}{Z_u r_k} \tag{8}$$

To obtain the final normalized coordinates for CPFG sampling, the component ratios are rescaled

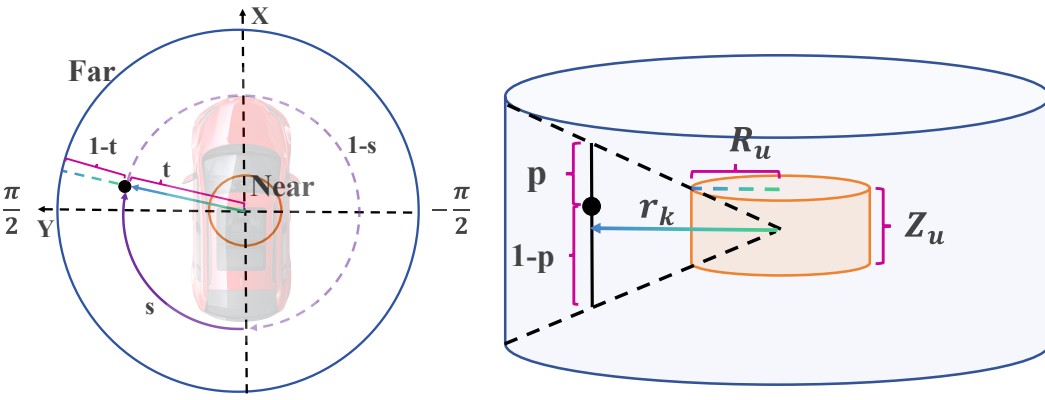

Figure 14: Proportional sampling in the radial and angular directions.

Figure 15: Proportional sampling in the vertical direction.

from the [0, 1] interval to the canonical range of [-1, 1]:

$$\mathbf{X}^k_n = (2t_k - 1, 2s_k - 1, 2p_k - 1)^T = (\frac{2(r_k - r_{min})}{r_{max} - r_{min}} - 1, \frac{-\theta_k}{\pi}, \frac{-2z_k R_u}{Z_u r})^T \tag{9}$$

Then, we can obtain the occ feature of $\mathbf{X}_o[;k]$ using the following equation:

$$\mathbf{f}_{occ}^k = \text{Trilinear}(\mathbf{F}^{occ}, \mathbf{X}_o[;k]) \tag{10}$$

Performing the above operation on each point, we can obtain $\mathbf{F}_{cy}^{occ}$ for the occupancy grid $\mathbf{O}$.

**Sampling Background Feature on the Cylinder Plane Feature.** In our cylindrical projection-based background rendering method, we need to compute ray direction vectors from the camera viewpoint for each pixel in the image. This process involves geometric transformations from pixel coordinates to the world coordinate system.

First, we transform pixel coordinates $(\mathbf{i}, \mathbf{j})$ in the image to normalized camera coordinates. Given the camera intrinsic matrix:

$$K = \begin{bmatrix} f_x & 0 & c_x \\ 0 & f_y & c_y \\ 0 & 0 & 1 \end{bmatrix} \tag{11}$$

where $f_x$, $f_y$ represent the focal lengths in the $x$ and $y$ directions respectively, and $(c_x, c_y)$ denotes the principal point coordinates. For any pixel position $(\mathbf{i}, \mathbf{j})$ in the image, its normalized coordinates $(x_c, y_c, 1)$ in the camera coordinate system can be computed as:

$$x_c = \frac{\mathbf{i} - c_x + 0.5}{f_x}, \quad y_c = \frac{\mathbf{j} - c_y + 0.5}{f_y}, \quad z_c = 1 \tag{12}$$

The $+0.5$ offset is used to convert pixel coordinates from the top-left origin to the pixel center, ensuring that rays emanate from the center of each pixel.

Next, we transform the direction vector $\mathbf{d}c = (x_c, y_c, 1)$ from the camera coordinate system to the world coordinate system. This transformation is achieved through the camera's rotation matrix $\mathbf{R}$.

Given the camera extrinsic matrix (camera-to-world transformation matrix):

$$\mathbf{T}_{c2w} = \begin{bmatrix} \mathbf{R} & \mathbf{t} \\ \mathbf{0}^T & 1 \end{bmatrix} \tag{13}$$

where $\mathbf{R}$ is a $3 \times 3$ orthogonal rotation matrix and $\mathbf{t}$ is the translation vector. The direction vector in the camera coordinate system is transformed to the world coordinate system as:

$$\mathbf{d}_w = \mathbf{R} \cdot \mathbf{d}_c = \mathbf{R} \cdot \begin{bmatrix} x_c \\ y_c \\ 1 \end{bmatrix} \tag{14}$$

Since the direction vector only represents direction, we typically normalize it to a unit vector:

$$\mathbf{d} = \frac{\mathbf{d}_w}{\|\mathbf{d}_w\|} = (d_x, d_y, d_z)^T \tag{15}$$

Finally, for camera position $\mathbf{p} = (x_0, y_0, z_0)$ and normalized direction vector $\mathbf{d}$, the ray can be parameterized as:

$$\mathbf{P}(t) = \mathbf{p} + t\mathbf{d}, \quad t \geq 0 \tag{16}$$

where parameter $t$ represents the distance along the ray direction. This parameterized ray equation will be used for subsequent intersection calculations with the background cylinder to determine the background texture sampling location for each pixel.

For a Cylinder Plane with radius $R_u$, the intersection point $\mathbf{P}(t) = \mathbf{p} + t\mathbf{d}$ must satisfy the following constraint:

$$(\mathbf{p} + t\mathbf{d})_x^2 + (\mathbf{p} + t\mathbf{d})_y^2 = R_u^2 \tag{17}$$

This can be rearranged into a linear equation with two variables:

$$t^2(d_x^2 + d_y^2) + 2t(x_0 d_x + y_0 d_y) + (x_0^2 + y_0^2 - R_u^2) = 0 \tag{18}$$

Solving for $t$, we get:

$$\hat{t} = \frac{-(x_0 d_x + y_0 d_y) + \sqrt{(x_0 d_x + y_0 d_y)^2 - (d_x^2 + d_y^2)(x_0^2 + y_0^2 - R_u^2)}}{d_x^2 + d_y^2} \tag{19}$$

The intersection point is therefore given by the coordinates:

$$P_{intersect} = \mathbf{p} + \hat{t} \cdot \mathbf{d} = (p_x, p_y, p_z)^T \tag{20}$$

To convert Cartesian coordinates to cylindrical coordinates, use the following formulas:

$$\alpha = \sqrt{p_x^2 + p_y^2}, \quad \theta = \arctan(p_y/p_x), \quad z = p_z \tag{21}$$

To generate sampling points suitable for interpolation, the cylindrical coordinates must be normalized. Since $\theta \in [-\pi, \pi)$, $z \in [-\frac{Z_u}{2}, \frac{Z_u}{2}]$, the mapping to a normalized space is performed as follows:

$$u = -\frac{\theta}{\pi}, \quad v = \frac{-z}{Z_u} \tag{22}$$

The normalized sampling coordinates for bilinear interpolation are therefore given by:

$$\mathbf{P}_{intersect} = (u, v)^T \tag{23}$$

**Pixel Projection Features.** Drawing inspiration from the feature projection method in Omni-Scene (Wei et al., 2025), we propose a novel projection paradigm for the CPFG representation. In contrast to Omni-Scene, which projects pixel features directly onto TPV (Huang et al., 2023) planes, our approach projects each pixel feature to its two nearest neighboring CPFG planes based on its spatial location. These projected features are then interpolated and fused using inverse distance weighting.

We transform the $k$-th pixel Gaussian centers with pixel features $\mathbf{f}_{pix}^k$, denoted as $p_{\text{pixel}}^k$, from the pixel branch into the CPFG space. For any point $\mathbf{X}_k = (r_k, \theta_k, z_k)$ within CPFG space, we compute the normalized sampling coordinates $(t_k, s_k, p_k)$ according to Eq. (6) and Eq. (8). Subsequently, using the radial component $t_k$ and the total number of CPFG planes, $K$, we identify the adjacent planes enclosing the sampling point. Specifically, the point is located between the CPFG planes with indices $\lfloor t_k \cdot K \rfloor$ and $\lfloor t_k \cdot K \rfloor + 1$. Finally, we calculate the precise relative position of the point between these two planes, which serves as the interpolation weight for subsequent operations. For the projected feature tensor $\mathbf{F}^{proj}$, which is initialized with zeros, we assign values to the corresponding positions according to Eq. (25).

$$\mathbf{F}^{proj}[\lfloor t_k \cdot K \rfloor, \lfloor s_k \rfloor, \lfloor p_k \rfloor] = (1 - q_k) \cdot \mathbf{f}_{pix}^k, \quad \mathbf{F}^{proj}[\lfloor t_k \cdot K \rfloor + 1, \lfloor s_k \rfloor, \lfloor p_k \rfloor] = q_k \cdot \mathbf{f}_{pix}^k$$
$$where \quad q_k = t_k \cdot K - \lfloor t_k \cdot K \rfloor \tag{24}$$

Subsequently, this projected feature $\mathbf{F}^{proj}$ is flipped and then added to both the clockwise and counter-clockwise volume features ($\mathbf{F}_{cy}^+$ and $\mathbf{F}_{cy}^-$), where $CV_{proj}$ is a convolution module:

$$\mathbf{F}_{cy}^+ = \mathbf{F}_{cy}^+ + CV_{proj}(\mathbf{F}^{proj})$$
$$\mathbf{F}_{cy}^- = \mathbf{F}_{cy}^- + CV_{proj}(\kappa(\mathbf{F}^{proj})) \tag{25}$$

## A.8 DATA PREPROCESSING

**Generated Sky Masks for the nuScenes Dataset.** For precise sky segmentation, we employed the LISA model by loading its pre-trained weights from the LISA-13B-llama2-v1-explanatory version. LISA is an advanced segmentation model fine-tuned from a Large Language Model (LLM), with the key advantage of comprehending natural language instructions. Accordingly, we utilized the text prompt "sky" to guide the model in automatically identifying and segmenting the sky regions within the images.

Fig. 16 illustrates a selection of visual results from our sky segmentation process. As can be clearly observed, the LISA model achieves excellent segmentation performance even under challenging low-light conditions, such as at night. Compared to conventional semantic segmentation models like SegFormer (Xie et al., 2021), LISA demonstrates significant advantages in both segmentation accuracy and robustness. We applied this segmentation procedure to all RGB images within the nuScenes dataset, encompassing both keyframes (samples) and intermediate frames (sweeps).

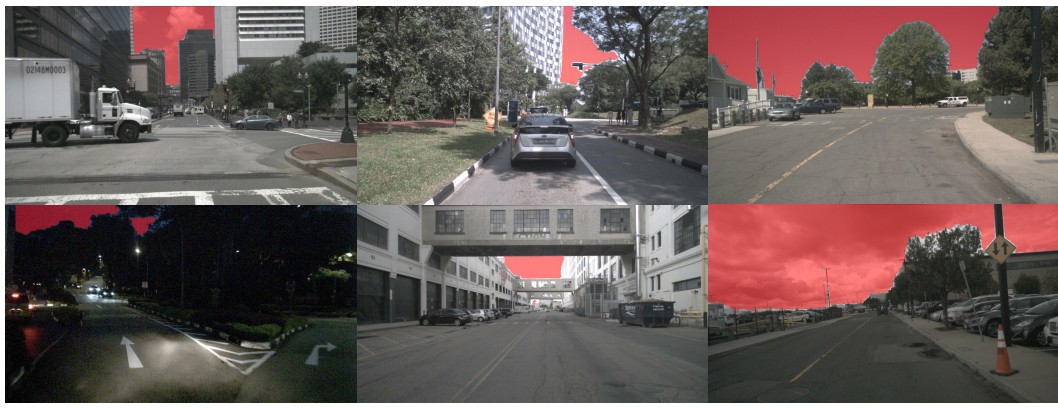

Figure 16: **Visualization of sky segmentation results from the LISA model on the nuScenes dataset.** Leveraging a language-guided segmentation model, we achieved highly accurate sky segmentation, even in challenging conditions such as complex and low-light scenes.

**Construction of a Custom nuScenes 3D Occupancy Dataset.** Referring to Wei et al. (2023), we utilize a comprehensive pipeline for generating dense 3D semantic occupancy grids from nuScenes LiDAR sequences. The method begins by initializing scene parameters including voxel size $\delta_v = 0.4$, occupancy grid dimensions $\Omega = [L_o, H_o, W_o] = [40, 200, 200]$, and point cloud range $\mathcal{R} = [-40, -40, -3, 40, 40, 13]$. For each scene, the system processes sequential LiDAR frames, extracting both static background points and dynamic objects.

Keyframe processing involves several stages. First, static points are aggregated across frames and transformed to a consistent coordinate system. Dynamic objects are handled through instance-centric point accumulation, where object points from multiple frames are transformed to canonical coordinates. The complete scene is reconstructed using Poisson surface reconstruction with configurable depth parameters and density-based vertex filtering, generating a watertight mesh $\mathcal{M}$.

The mesh is then voxelized within the predefined spatial bounds $\mathcal{R}$, converting continuous surfaces into discrete 3D grids. Semantic labels are propagated to voxels through nearest-neighbor matching using the Chamfer distance between voxel centers and annotated sparse points. Finally, a denoising stage applies connected-component filtering to remove small noise regions, 3D morphological operations to smooth boundaries, and flood-fill algorithms to complete internal cavities while preserving semantic consistency.

The pipeline efficiently handles large-scale driving scenes through temporal aggregation of static elements and object-centric processing of dynamic entities. Output occupancy grids preserve fine-grained scene structures with accurate semantic labeling, suitable for downstream autonomous driving perception tasks. Memory management includes explicit garbage collection after scene processing to maintain computational efficiency.

Fig. 17 illustrates visualization examples from the Occ dataset, which contains 17 semantic classes distinguished by different colors. In our methodology, we merge all non-air classes into a single foreground class, thereby simplifying the original multi-class task into a binary classification problem for supervision.

**Construction of the Carla-Centric Dataset.** We adopted and extended SEED4D (Kästingschäfer et al., 2025), a synthetic data generation system based on the CARLA simulator (Dosovitskiy et al., 2017), which is designed to create dynamic driving scenarios with ego-inward views. To ensure our generated data fully aligns with the format and specifications of the nuScenes benchmark dataset, we made three key modifications to the original SEED4D system. First, we adjusted the camera configuration to match that of nuScenes, deploying five surround-view cameras with a 70-degree Field of View (FoV) and one rear-view camera with a 110-degree FoV. Second, we enabled the rendering of the ego-vehicle, ensuring it is visible in both ego-inward perspectives. Third, we integrated an additional occupancy grid generation module to synthesize precise, scene-aligned occupancy labels, with its configuration also adhering to the nuScenes standard.

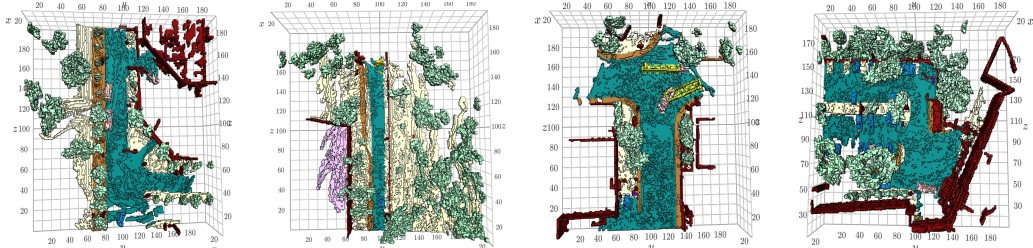

Figure 17: **Visualization of nuScenes occupancy labels generated from semantic point cloud processing.** Due to the limited precision of real-world data acquisition and errors from semantic point cloud processing, the occupancy representation in the nuScenes dataset exhibits a considerable amount of noise artifacts and sharp geometric features.

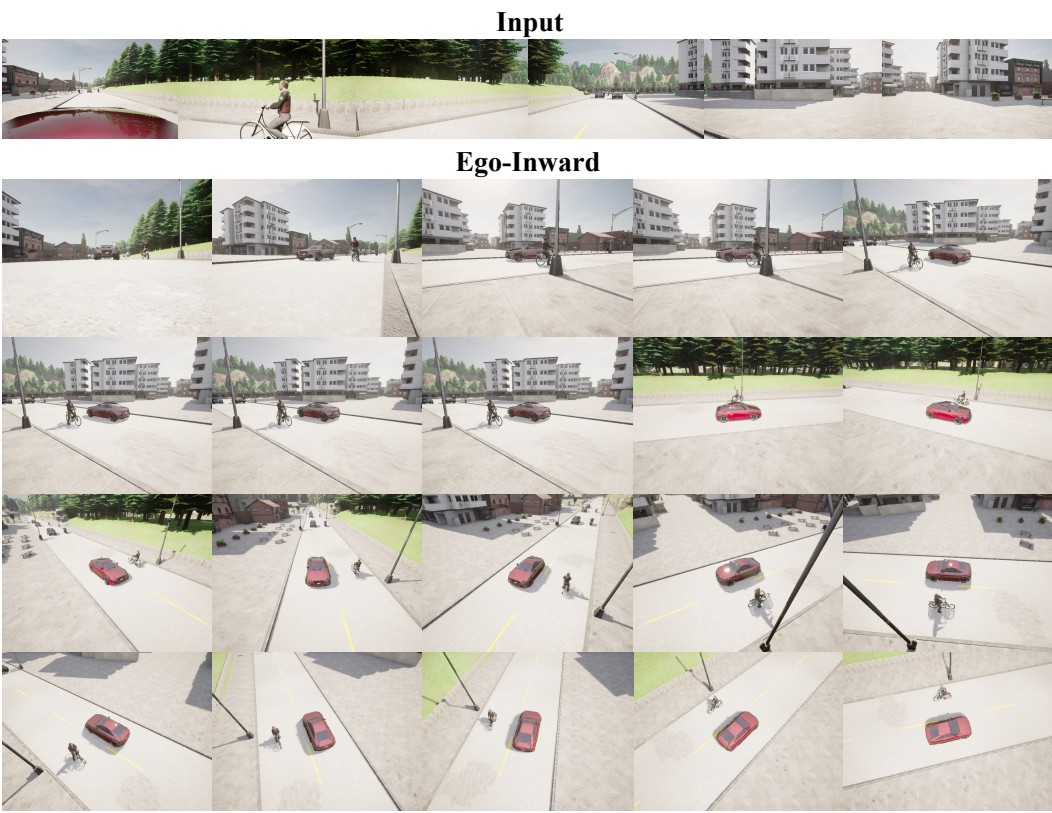

Figure 18: **A visualization example of a scene from the Carla-Centric dataset.** We present a set of six outward-looking images from the ego vehicle, along with 20 images sampled from a larger collection of 100 views that are uniformly distributed on a hemisphere and oriented towards the scene center.

Fig. 18 illustrates a typical scene from our constructed Carla-Centric dataset. Each scene consists of 6 input images and 100 ego-inward images. For clarity, only a subset of 20 ego-inward images from this scene is presented in the figure.

Fig. 19 showcases the visualization of occupancy labels from our self-developed Carla-Centric dataset. In contrast to datasets based on real-world sensor data like nuScenes, our approach leverages the near-perfect ground truth available from the CARLA simulation environment. Benefiting from this high-fidelity ground truth, the resulting aggregated occupancy grids exhibit significant smoothness and spatial consistency, effectively mitigating the noise and sparsity issues commonly found in real-world data.

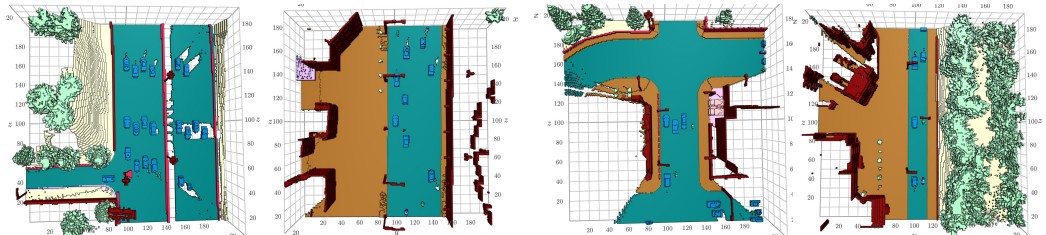

Figure 19: **The visualization of our proposed Carla-centric 3D occupancy.** In stark contrast to the results from nuScenes, our method generates a significantly smoother geometry that is free from noise artifacts and topological holes.

**Selection of the Feature Extractor.** To identify the optimal feature extractor, we conducted a comparative analysis of several mainstream models as Fig. 20 shows. Although the DINO series models (e.g., DINOv2 (Oquab et al., 2024)) demonstrate powerful feature extraction capabilities, they are known to have certain issues, such as the presence of artifacts in the feature maps generated by DINOv2. Consequently, we focused our investigation on fine-tuned variants of DINO (Caron et al., 2021), primarily examining FeatUp (Fu et al., 2024) and FiT3D (Yue et al., 2024). Specifically, FeatUp addresses the low-resolution issue of the original DINO features, while FiT3D is optimized for 3D Gaussian Splatting (3DGS) scenes, making it more suitable for 3D tasks. Furthermore, for additional comparative visualization, we also utilized features extracted by the open-source Radio-v2.5 (Heinrich et al., 2024) model from NVIDIA as a supplementary baseline.

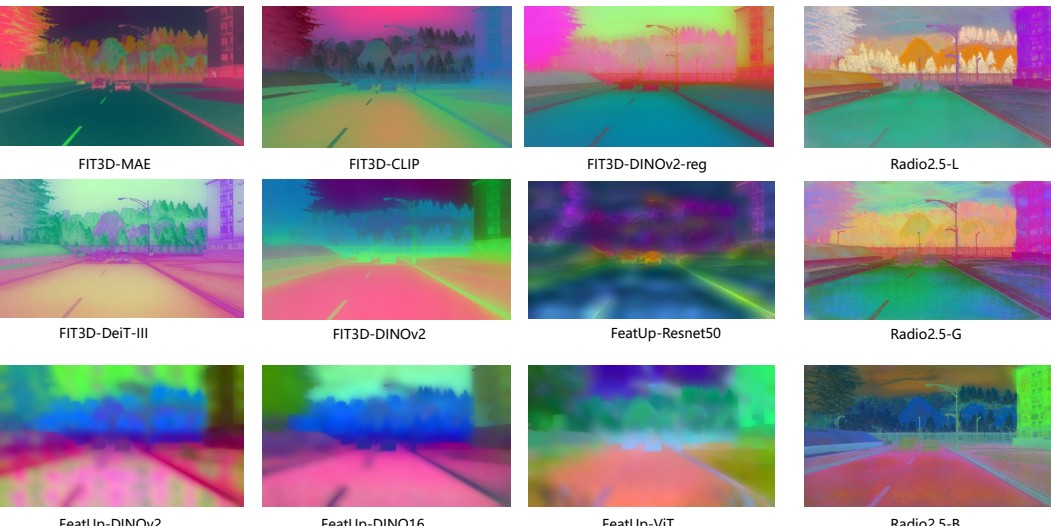

Figure 20: **Comparative visualization of feature maps extracted by different backbones.** Different families of feature extractors demonstrate distinct characteristics, which in turn lead to a significant disparity in performance.

To assess the representation quality of different models, we employed Principal Component Analysis (PCA) to visualize their feature maps. The analysis revealed that models fine-tuned with FiT3D, regardless of the backbone, produced overly smooth feature maps, potentially leading to a loss of fine-grained details. Meanwhile, models fine-tuned with FeatUp consistently exhibited severe artifacts and blurring, which compromised feature reliability. In stark contrast, the Radio-v2.5 series demonstrated a superior capability in preserving feature details and maintaining strong spatial consistency, an observation that aligns with the high praise it received in Feat2GS (Chen et al., 2025). Therefore, based on this qualitative analysis and its clear advantages in feature quality, we ultimately selected Radio-v2.5-B as the feature extractor for our method.

## A.9 IMPLEMENT DETAILS

### A.9.1 OVERVIEW OF EVALUATION METRICS.

**Peak Signal-to-Noise Ratio (PSNR).** A widely used metric for quantifying the reconstruction quality of lossy compression and generation tasks. It measures the ratio between the maximum possible power of a signal and the power of corrupting noise that affects its fidelity. For an 8-bit image with a maximum possible pixel value of $L = 255$ and a size of $H \times W$, the PSNR between a ground-truth image $I$ and a reconstructed image $\hat{I}$ is defined in decibels (dB) as:

$$\text{PSNR}(I, \hat{I}) = 10 \cdot \log_{10} \left( \frac{L^2}{\text{MSE}(I, \hat{I})} \right)$$

Where the Mean Squared Error (MSE) is calculated as $\text{MSE}(I, \hat{I}) = \frac{1}{HW} \sum_{i=1}^{H} \sum_{j=1}^{W} (I_{ij} - \hat{I}_{ij})^2$. A higher PSNR value indicates a lower level of error, signifying that the reconstructed image is closer to the original. While simple and computationally efficient, PSNR's reliance on pixel-wise differences means it may not always align perfectly with human perceptual judgment of image quality.

**Structural Similarity Index Measure (SSIM).** A perceptual metric designed to better approximate the human visual system's assessment of image similarity. Unlike PSNR, SSIM evaluates the degradation of quality as a change in structural information. It compares two images, $I$ and $\hat{I}$, based on three components: luminance ($l$), contrast ($c$), and structure ($s$). For two image patches $x$ and $y$ from $I$ and $\hat{I}$ respectively, the SSIM is computed as:

$$\text{SSIM}(x, y) = \frac{(2\mu_x \mu_y + c_1)(2\sigma_{xy} + c_2)}{(\mu_x^2 + \mu_y^2 + c_1)(\sigma_x^2 + \sigma_y^2 + c_2)}$$

Where $\mu_x, \mu_y$ are the local means, $\sigma_x, \sigma_y$ are the local standard deviations, and $\sigma_{xy}$ is the cross-covariance. The constants $c_1 = (k_1 L)^2$ and $c_2 = (k_2 L)^2$ are included to stabilize the division. The final SSIM score is the mean of the SSIM values computed over all local windows in the image. The score ranges from -1 to 1, where 1 indicates perfect structural similarity.

**Learned Perceptual Image Patch Similarity (LPIPS).** Also known as "perceptual loss", it measures the distance between two images in a perceptually relevant feature space. It more closely mirrors human perception of image similarity than traditional metrics like PSNR and SSIM. To compute LPIPS, two images, $I$ and $\hat{I}$, are passed through a pre-trained deep neural network (e.g., VGG or AlexNet). The feature activations are extracted from multiple layers, $l$. For each layer, the activations are unit-normalized in the channel dimension ($F^l, \hat{F}^l \in \mathbb{R}^{H_l \times W_l \times C_l}$). The L2 distance is then computed, scaled by a learned weight vector $w_l$, and averaged over spatial dimensions ($H_l, W_l$):

$$\text{LPIPS}(I, \hat{I}) = \sum_l \frac{1}{H_l W_l} \sum_{h,w} \|w_l \odot (F^l_{hw} - \hat{F}^l_{hw})\|_2^2$$

The final LPIPS score is the sum of distances across all considered layers. A lower LPIPS score signifies that the two images are more similar from a perceptual standpoint, indicating a higher-quality reconstruction.

**Pearson Correlation Coefficient (PCC).** A statistical measure that evaluates the linear relationship between two sets of data. In the context of computer vision, it is often applied to depth map evaluation, where it assesses the correlation between the predicted depth values and the ground-truth depth values, irrespective of absolute scale and shift. For a predicted depth map $\hat{D}$ and a ground-truth depth map $D$, the PCC is defined as the covariance of the two variables divided by the product of their standard deviations:

$$\text{PCC}(D, \hat{D}) = \frac{\text{cov}(D, \hat{D})}{\sigma_D \sigma_{\hat{D}}} = \frac{\sum_{i=1}^{N}(D_i - \mu_D)(\hat{D}_i - \mu_{\hat{D}})}{\sqrt{\sum_{i=1}^{N}(D_i - \mu_D)^2}\sqrt{\sum_{i=1}^{N}(\hat{D}_i - \mu_{\hat{D}})^2}}$$

Where $N$ is the total number of valid pixels, $\mu_D$ and $\mu_{\hat{D}}$ are the mean depth values, and $\sigma_D$ and $\sigma_{\hat{D}}$ are their standard deviations. The PCC ranges from -1 to +1, where +1 indicates a perfect positive linear correlation, 0 indicates no linear correlation, and -1 indicates a perfect negative linear correlation. For depth evaluation, a higher PCC value is desirable.

### A.9.2 A BRIEF OVERVIEW OF THE BASELINES.

**6Img-to-3D.** We utilize the official implementation of 6Img-to-3D[1]. Diverging from the original seed4d dataset employed in their work, we construct a dataset that is fully aligned with nuScenes. Please refer to Appendix A.6 for detailed distinctions between these datasets. Additionally, we integrate PCC evaluation metrics into the codebase to facilitate comprehensive experimental comparisons.

**Omni-Scene.** We utilize the official repository[2] of Omni-Scene. For the Carla-Centric dataset, we adjust the computation of ray origins and direction vectors to accommodate its distinct coordinate system. To align with the dataset resolution of nuScenes as used in Omni-Scene, we downsampled the Carla-Centric input and target images by a factor of four. For the experiments on nuScenes, we kept the settings identical to theirs.

We assign the maximum depth value to sky regions, with an associated confidence of 1.0. To maintain a rigorous and fair evaluation, the rendered depth maps undergo a filtering step at inference time: any depth value surpassing a predefined threshold is subsequently set to zero, thereby ensuring the consistency of the depth map in our model.

**DrivingForward.** We utilize the official repository[3] of DrivingForward. The original DrivingForward architecture is designed to leverage contextual information from three consecutive frames (t-1, t, t+1) to achieve high-quality reconstruction. For a fair comparison with our single-frame approach, we adapted its single-frame (SF) variant for our experiments. Furthermore, we addressed several underlying issues within its codebase to ensure compatibility with our dataset. A key modification was replacing its rendering kernel to enable the accurate generation of depth maps, a prerequisite for computing the Pearson Correlation Coefficient (PCC) metric.

**Depthsplat & MVSplat & PixelSplat.** We utilize the official repositories of Depthsplat[4], MVSplat[5] and PixelSplat[6]. For the Carla-Centric dataset, which provides ground-truth (GT) depth maps, we directly utilize these for supervision, bypassing the need for pseudo-labels from models like DepthAnything. To ensure the optimal performance of baseline models that require specific input aspect ratios, we maintain their original image resolutions during their respective training and inference processes. For a fair and consistent evaluation, the final outputs from all methods are rendered to a common resolution matching that of our model before comparison.

**SplatterImage.** We utilize the official repository[7] of SplatterImage. We observe that the baseline model suffers from significant training instability when applied to our dataset. The training process is prone to divergence, often collapsing after approximately 3,000 iterations. This instability is highly sensitive to the random initialization seed, frequently resulting in catastrophic performance with PSNR values below 10. To establish a meaningful benchmark for comparison, we conduct an extensive search across dozens of random seeds and report the results from the most successful run.

### A.10 SEARCH FOR THE MOST SUITABLE HYPERPARAMETERS

We conduct an ablation study to analyze the impact of two critical hyperparameters on reconstruction quality: the number of CPFG feature groups, K, and the number of predicted Gaussians per voxel in the volume-aware branch, $G_v$. We evaluated combinations of K from 36, 48, 60 and $G_v$ from 1, 2, 3, resulting in nine experimental configurations per dataset.

As shown in Fig. 21 and Fig. 22, on the nuScenes dataset, the model achieves its best PSNR with the configuration ($K = 48, G_v = 3$). The PSNR is notably sensitive to the value of $G_v$; performance improves as $G_v$ increases. This is because models trained on nuScenes are challenged by high-frequency texture details, which cause the pixel branch to produce holes or voids. This, in turn,

---

[1]https://github.com/continental/6Img-to-3D

[2]https://github.com/WU-CVGL/Omni-Scene

[3]https://github.com/fangzhou2000/DrivingForward

[4]https://github.com/cvg/depthsplat

[5]https://github.com/donydchen/mvsplat

[6]https://github.com/dcharatan/pixelsplat

[7]https://github.com/szymanowiczs/splatter-image

Table 4: **Quantitative comparison of our model against the baselines.** The best, second-best, and third-best results are marked with corresponding cell colors, reflecting nuanced performance across different metrics.

| | | Carla-Centric | | | | nuScenes | | | |
|---|---|---|---|---|---|---|---|---|---|
| $K$ | $G_v$ | PSNR↑ | LPIPS↓ | SSIM↑ | PCC↑ | PSNR↑ | LPIPS↓ | SSIM↑ | PCC↑ |
| 36 | 1 | 17.61 | 0.448 | 0.582 | 0.796 | 24.01 | 0.275 | 0.705 | 0.853 |
| 36 | 2 | 17.58 | 0.451 | 0.579 | 0.794 | 24.16 | 0.268 | 0.712 | 0.859 |
| 36 | 3 | 17.63 | 0.449 | 0.581 | 0.795 | 24.28 | 0.262 | 0.719 | 0.866 |
| 48 | 1 | 18.38 | 0.362 | 0.619 | 0.818 | 24.71 | 0.250 | 0.733 | 0.871 |
| 48 | 2 | 18.37 | 0.364 | 0.620 | 0.819 | 24.92 | 0.244 | 0.755 | 0.889 |
| 48 | 3 | 18.40 | 0.359 | 0.622 | 0.817 | 24.97 | 0.231 | 0.750 | 0.887 |
| 60 | 1 | 18.28 | 0.375 | 0.611 | 0.809 | 24.46 | 0.255 | 0.729 | 0.868 |
| 60 | 2 | 18.42 | 0.368 | 0.616 | 0.814 | 24.68 | 0.246 | 0.746 | 0.880 |
| 60 | 3 | 18.35 | 0.371 | 0.615 | 0.812 | 24.65 | 0.237 | 0.742 | 0.875 |

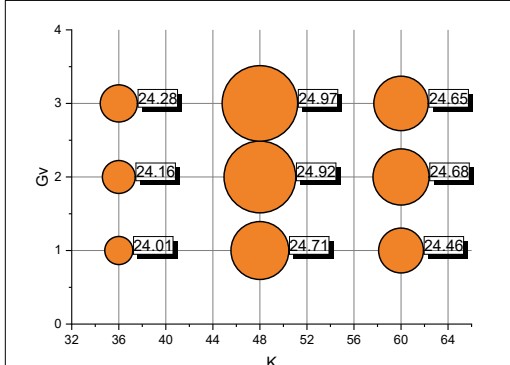 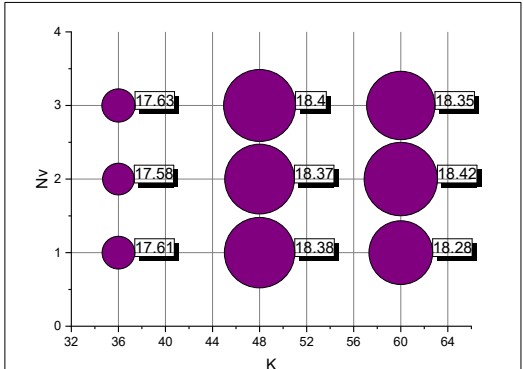

Figure 21: PSNR on the nuScenes dataset for models trained from scratch with varying hyperparameters.

Figure 22: PSNR on the Carla-Centric dataset for models trained from scratch with varying hyperparameters.

requires the volumetric representation branch to fill these voids as much as possible. A smaller $G_v$ value weakens this gap-filling capability, leading to lower performance.

In contrast, for the Carla-Centric dataset, the model demonstrates greater sensitivity to the value of $K$. This stems from its ego-inward setting, which demands high fidelity across all parts of the scene, unlike the limited forward-facing perspective. This makes the volumetric representation dominant. Consequently, a small $K$ leads to low field resolution and thus an inaccurate volumetric representation. Conversely, while a larger K might improve performance, it comes at the cost of significantly higher GPU memory consumption.

Taking into account the performance on both datasets, the frequency of peak performance occurrences in the ablation table, and the inherent trade-offs, we select $K = 48, G_v = 3$ as our optimal configuration.

## A.11    MORE EXPERIMENTAL RESULTS

**Visualization of the Prediction Results from the Occupancy-Aware Branch.** Fig. 23 visualizes the results from the occupancy-aware (occ-aware) branch of our model. This branch takes six-view images as input and outputs a classification score for each voxel in the 3D space. A voxel is classified as occupied if its occupancy score surpasses its "air" score.

Compared with the ground truth (GT) occupancy, the geometry predicted by our model is considerably smoother and more regular. This is attributed to the model's strong generalization capability, which effectively filters out the significant noise present in the GT labels. This effect is particularly evident in the geometry of "trees". Due to the high uncertainty inherent in the complex structure of dense branches and leaves, the model tends to learn and predict a smooth, "averaged" overall geometry, rather than a fine-grained structure with numerous internal voids. Furthermore, we also present the Bird's-Eye-View (BEV) feature map from this branch. As can be clearly observed, the feature

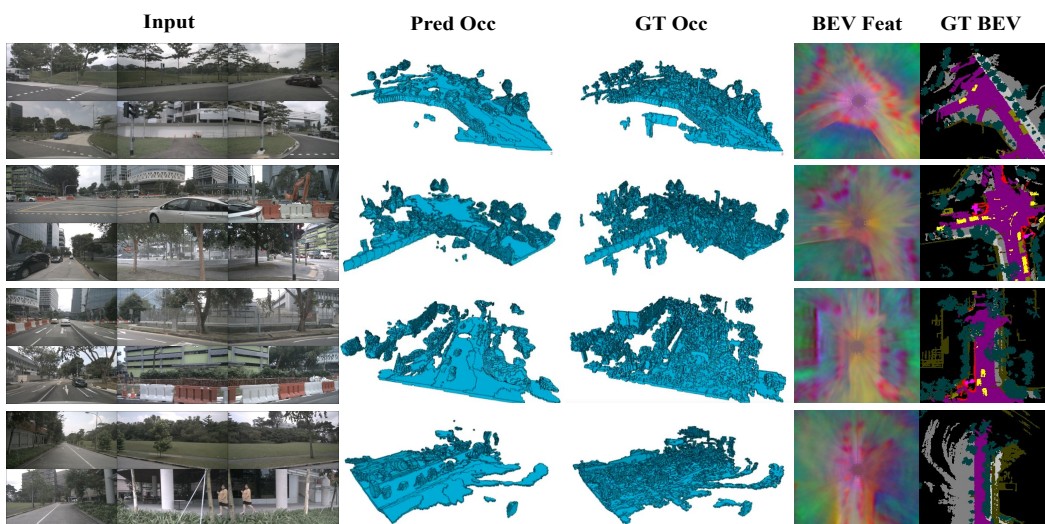

Figure 23: **Visualization of occupancy prediction results on the nuScenes dataset.** Furthermore, we present visualizations of the features from our BEV branch. In contrast to the Ground Truth, the predicted occupancy exhibits superior smoothness. We attribute this improvement to the strong generalization capability of our feedforward architecture, which enhances the model's robustness against inherent noise within the dataset. The BEV feature maps also demonstrate strong semantic discriminability, particularly for the "tree" category.

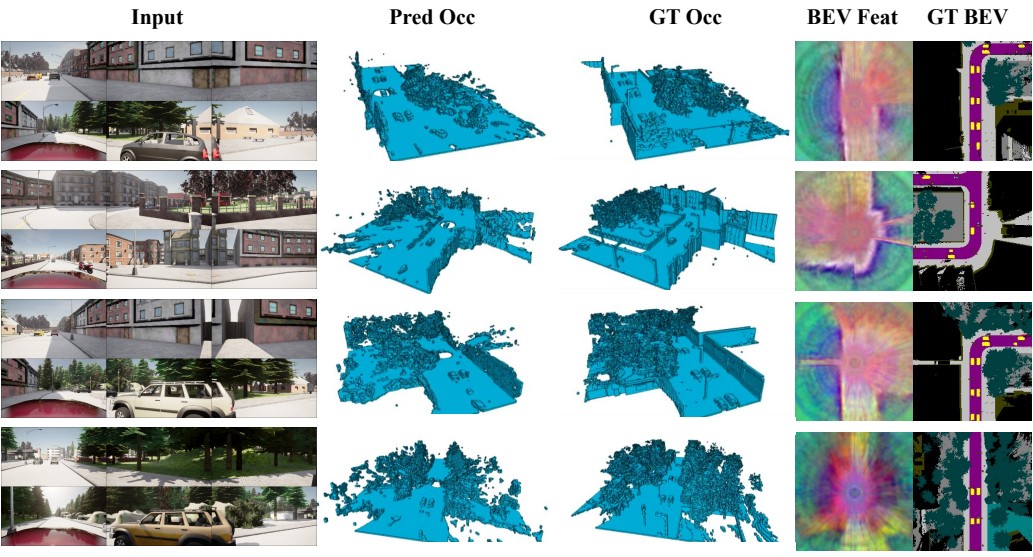

Figure 24: **Qualitative results on sample frames from the nuScenes dataset.** As demonstrated, the model produces high-fidelity predictions, where the BEV features exhibit clear semantic discriminability. This is evident in the distinct separation between classes such as "wall" and "drivable surface".

response in the regions corresponding to trees is highly prominent and exhibits a consistent color. This indicates that our model has successfully learned a common and robust feature representation for the tree category.

Fig. 24 visualizes the occupancy-branch predictions trained on Carla-Centric synthetic data. Thanks to the absence of sensor noise in simulation, the occupancy labels are highly accurate, and the scenes

are less complex than those in real-world nuScenes. Consequently, the model achieves superior predictive performance.

**More Visualization of Models Trained on Carla-Centric.** Fig. 25 presents additional qualitative results on the Carla-centric dataset to further demonstrate the superiority of our method. Our model not only synthesizes novel-view images with richer high-frequency details but also produces depth maps that more accurately align with the ground truth. In stark contrast, the baseline methods suffer from noticeable blurring artifacts and significant voids (holes) in their rendered images and depth estimations.

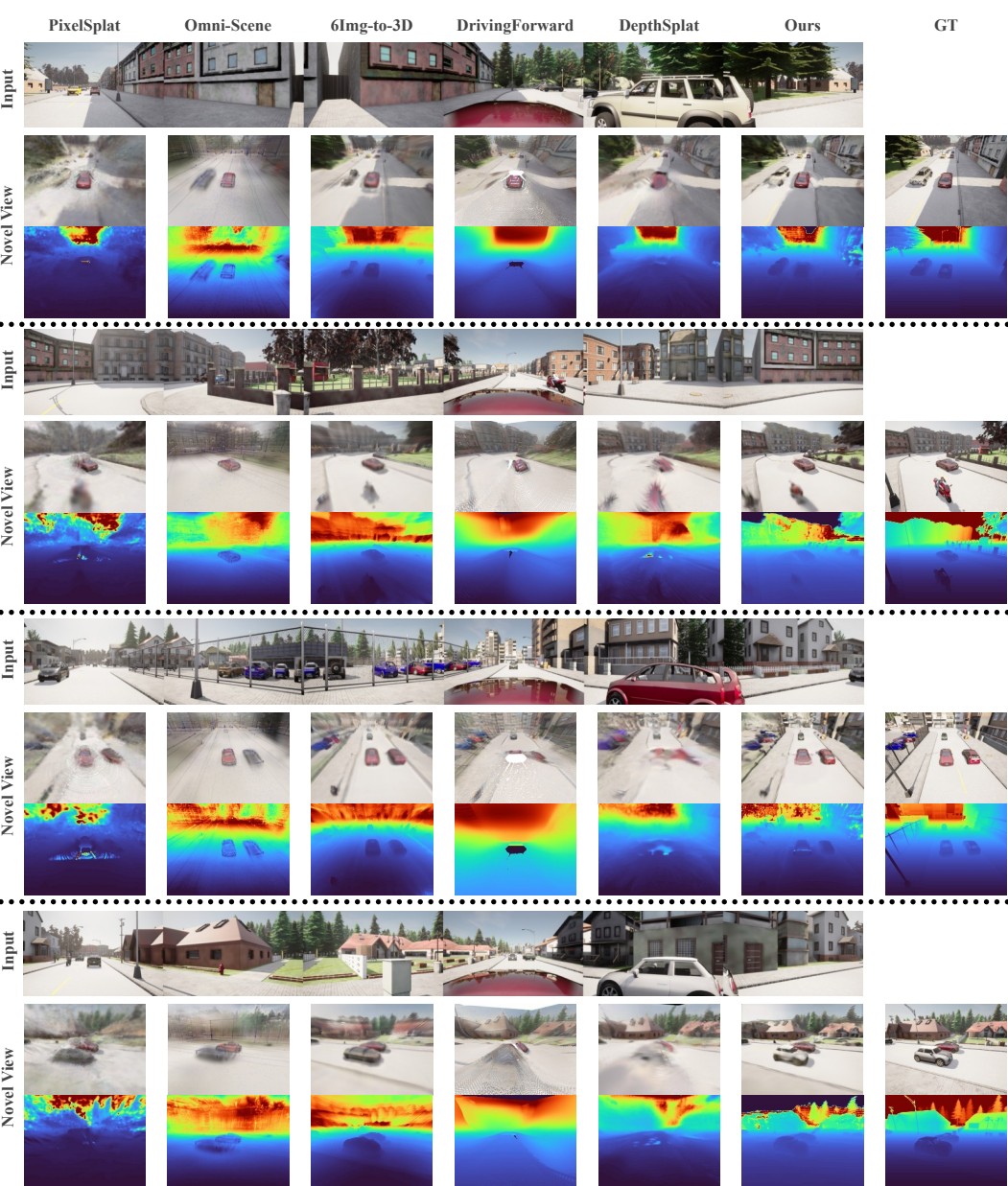

Figure 25: **Qualitative results on the Carla-Centric dataset.** Our model demonstrates superior performance in generating both photorealistic imagery and accurate geometry. It produces images with significantly sharper texture details and concurrently estimates highly accurate depth maps. Notably, our method effectively eliminates the blurring artifacts and voids (holes) that commonly plague other approaches, resulting in visually clean and structurally complete scene representations.

**More Visualization of Models Trained on nuScenes.** As visualized in Fig. 26, our model demonstrates a remarkable ability to reconstruct scenes with both high photometric realism and strong geometric integrity on the nuScenes dataset. In comparison, baseline methods fail on both fronts, suffering from a range of issues such as blurred textures, color shifts, distorted geometry, and unreliable depth. Our approach, conversely, excels in all these aspects, producing sharp textures, accurate colors, and a structurally sound 3D representation, as confirmed by its highly accurate depth maps.

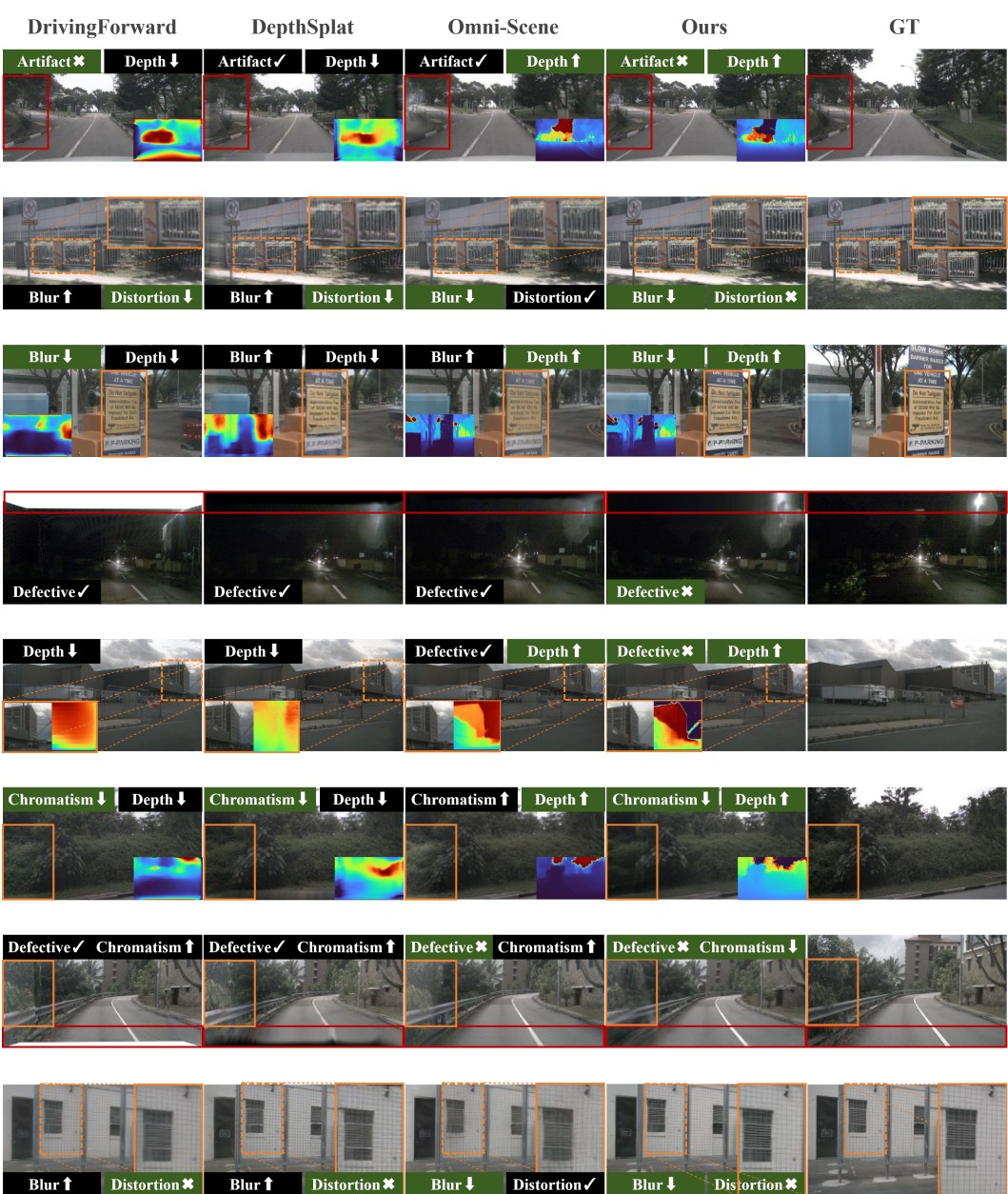

Figure 26: **The visualization results on the nuScenes dataset.** Our model achieves smaller photometric error and superior geometric representation. Particularly for high-frequency details like fences, our model renders fine textures without significant distortion. Furthermore, our model is less prone to artifacts and demonstrates better clarity on certain textures, such as text. Additionally, thanks to our sky-decoupled representation, we obtain a more complete geometry. In contrast, baseline methods tend to incorrectly learn distant, detailed objects into the sky's depth.

**More Zero-shot Visualization.** We use the model trained on nuScenes to test its zero-shot capability on the Waymo dataset. It is worth noting that nuScenes employs a 360-degree surround-view setup with a total of six cameras: five with a 70-degree Field of View (FOV) and one rear camera with a 110-degree FOV. All six cameras share the same resolution. In contrast, the Waymo dataset has only five cameras, covering a scene of slightly more than 180 degrees, and there are resolution differences between the cameras. The Pandaset and ONCE datasets are similar to nuScenes, both featuring six cameras and a 360-degree view, but their camera intrinsics differ. The Argoverse dataset contains seven cameras, also covering a 360-degree scene. The camera configurations for these datasets are summarized in Table 5.

We conduct our experiments using only the images captured by the surround-view cameras and modify the relevant parameters of UCCM for each dataset, as shown in Table 6. Special handling is required for the Waymo dataset due to the severe lack of viewing angles.

Table 5: Summary of camera configurations for common autonomous driving datasets.

| Dataset | Number of cameras | Coverage | Horizontal Field of View |
|---|---|---|---|
| nuScenes | 6 ring cameras | =360° | 70°,70°,70°,110°,70°,70° |
| Waymo | 5 ring cameras | >180° | 50°,50°,50°,50°,50° |
| Pandaset | 6 ring cameras | =360° | 50°,107°,107°,107°,107°,107° |
| ONCE | 6 ring cameras + 1 wide-angle camera | =360° | 90°,90°,90°,90°,90°,90° |
| Argoverse | 7 ring cameras + 2 stereo cameras | =360° | 69°,69°,69°,69°,69°,69°,69° |

Table 6: UCCM Parameter Settings for Zero-shot Generalization on Other Autonomous Driving Datasets.

| Dataset | UCCM parameters | | | | | |
|---|---|---|---|---|---|---|
| | $\rho_o$ | $\Delta h_o$ | $\rho_v$ | $\Delta h_v$ | $\rho_p$ | $\Delta h_p$ |
| nuScenes | 0.90 | 0.00 | 0.98 | 0.40 | 0.98 | 0.40 |
| Waymo | 1.20 | 0.00 | 0.98 | 0.40 | 0.98 | 0.40 |
| Pandaset | 2.00 | 0.00 | 1.60 | 0.00 | 1.60 | 0.00 |
| ONCE | 1.80 | 0.00 | 1.00 | 0.50 | 1.00 | 0.50 |
| Argoverse | 0.09 | 0.00 | 0.98 | 0.00 | 0.98 | 0.00 |

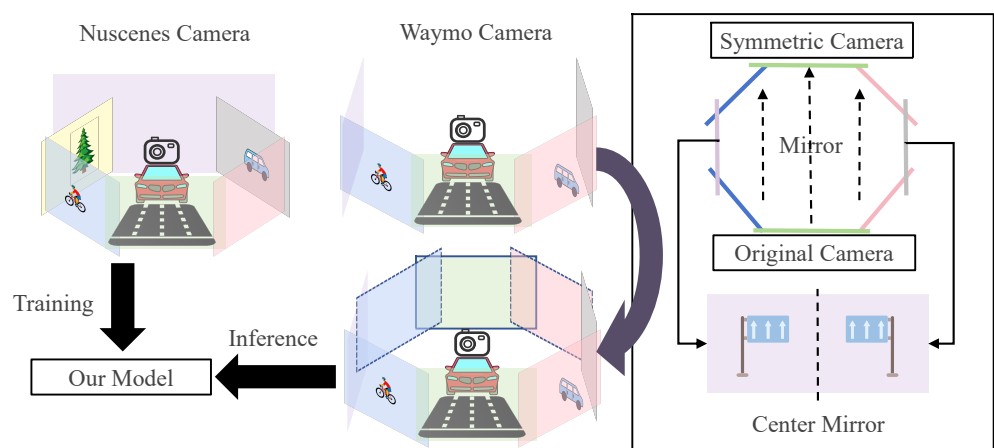

Figure 27: **Processing method for Waymo cameras.** Symmetric virtual cameras are constructed to complete unseen views and ensure pixel-level continuity. This process allows models trained on nuScenes to generalize directly to the Waymo dataset.

For the Waymo dataset, we adopt the processing method illustrated in Fig. 27. Specifically, we mirror the front, front-left, and front-right cameras across a plane to the rear of the vehicle, thereby

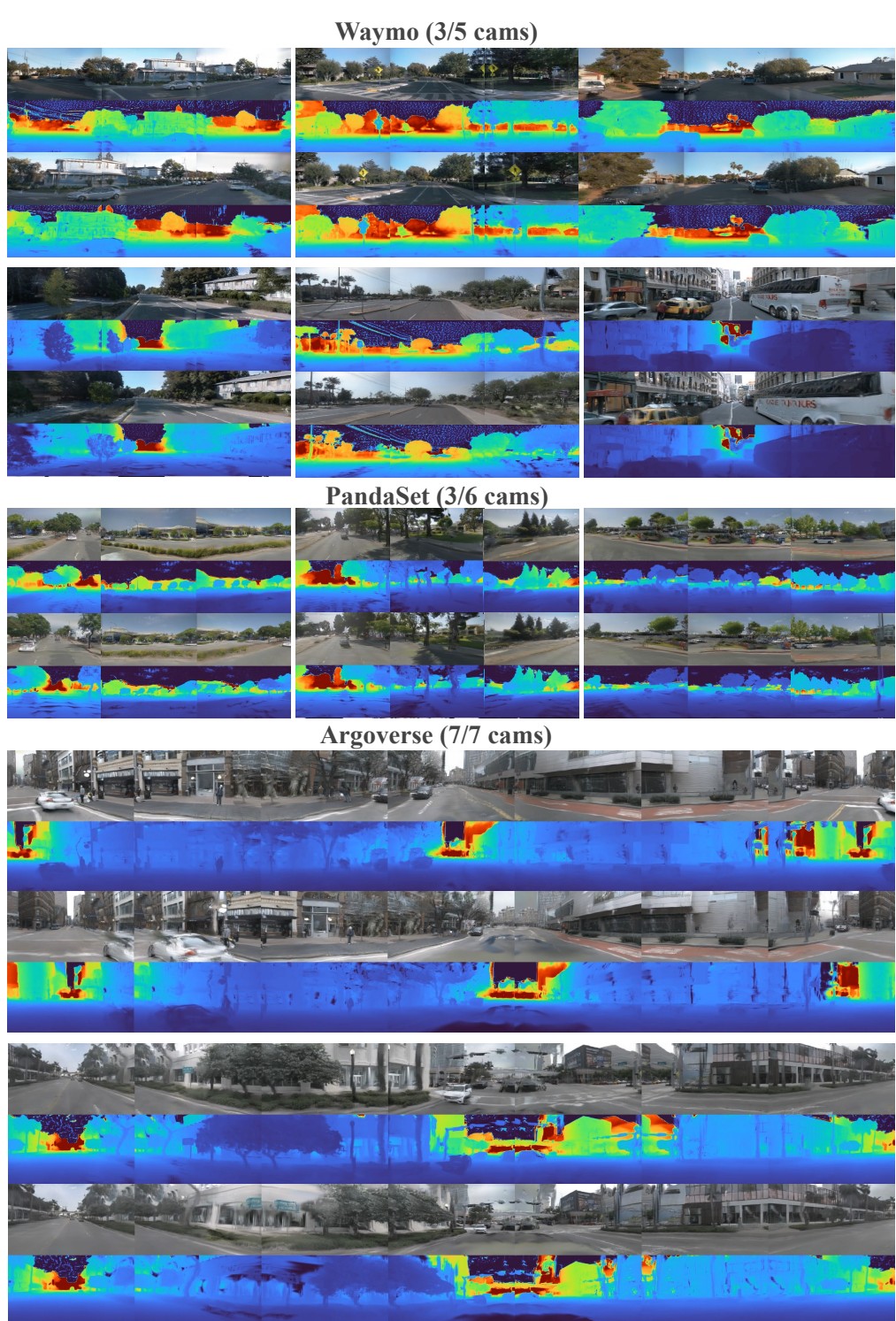

Figure 28: **More zero-shot visualization results.** The model trained on nuScenes generalizes well to other datasets.

constructing three symmetric virtual cameras. The images for these virtual cameras are mirrored versions of the forward-facing ones. For the side cameras, we horizontally flip their images along the vertical centerline to create virtual side cameras. This process results in an 8-camera configuration

including the virtual cameras, whose collective views are seamlessly connected from left to right. Fig. 28 showcases the strong camera adaptability of our model with additional zero-shot results on the Waymo, Pandaset, and Argoverse datasets.

## A.12   MORE APPLICATION RESULTS

**Integrate with Generative Models.** As a model designed for lifting 2D driving scenes to 3D, our approach exhibits a natural compatibility with 2D generative models. To demonstrate this, we have selected MagicDrive-V2 (Gao et al., 2024b), a classic model in driving scene video generation, to enable the generation of 3D driving scenes from text prompts.

As illustrated in Fig. 29, our model can serve as a powerful interface for state-of-the-art 2D generative models. We demonstrate this by integrating it with MagicDrive-V2 for text-to-3D driving scene generation. In this workflow, our model is tasked with establishing a robust and geometrically accurate 3D scene structure, which MagicDrive-V2 then "paints" with photorealistic textures guided by a text prompt. This synergy highlights that our method effectively bridges the gap from 2D generation to 3D-consistent scene creation, empowering existing models with strong 3D control.

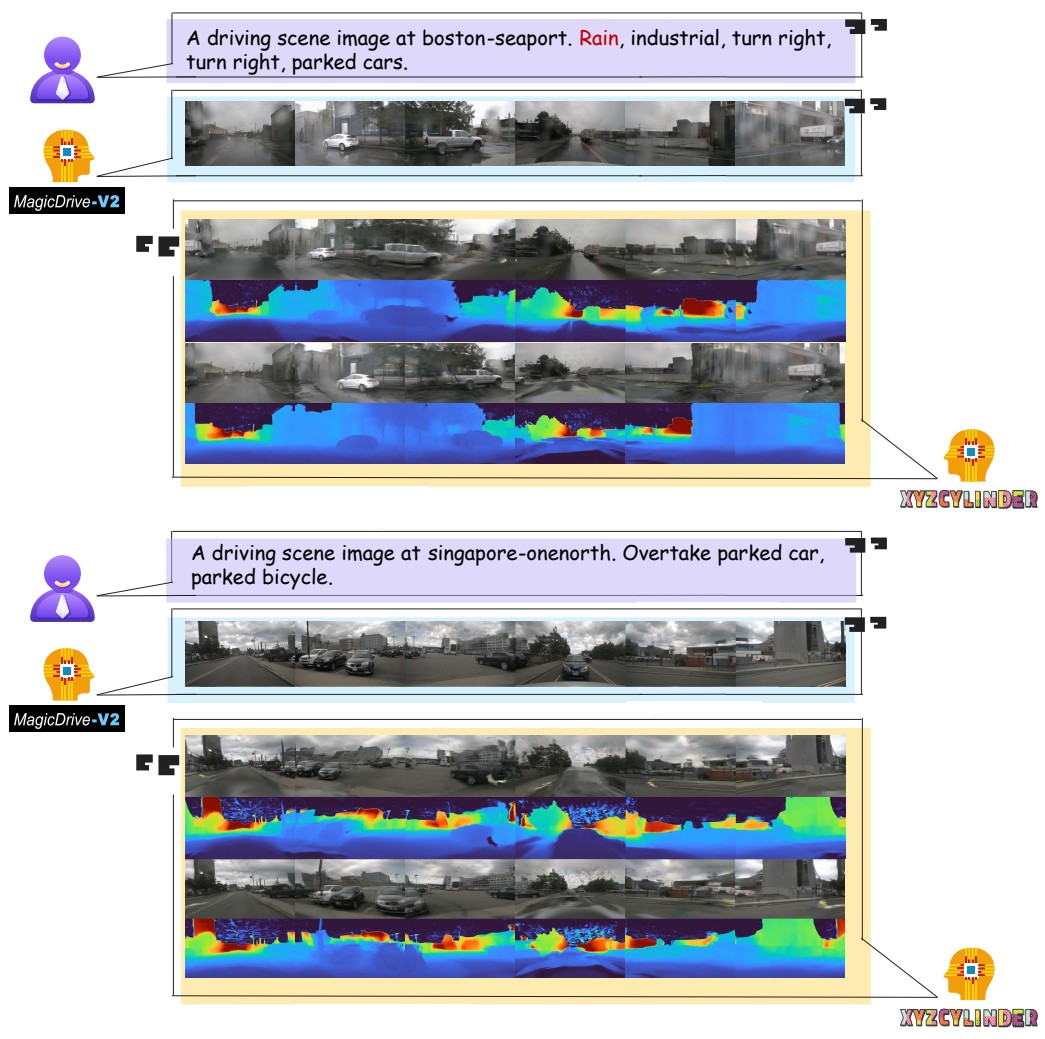

Figure 29: **Application cases 1:** Combining our model with advanced driving scene generators with text prompts.

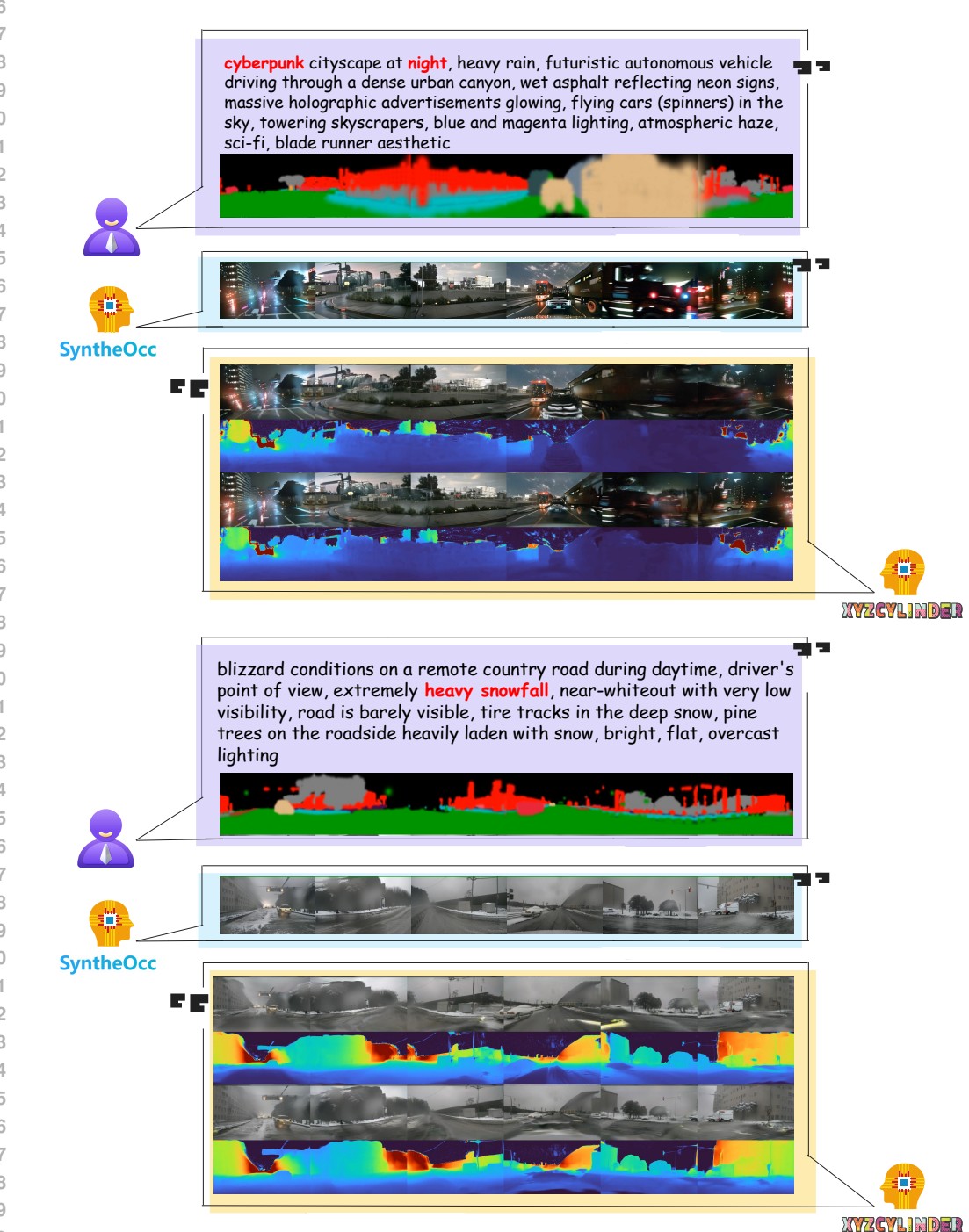

Figure 30: **Application cases 2:** Combining our model with advanced driving scene generators with text prompts and occupancy conditions.

To further enhance the controllability of our generation process, we demonstrate that our model can be seamlessly integrated with layout-based generative models. We chose SyntheOcc (Li et al., 2024a), a voxel-conditioned scene generator, and fine-tuned our model to incorporate its Gaussian-rendered semantic occupancy as an additional control signal. This hybrid approach enables a powerful decoupling of control: users can define the high-level artistic style and global attributes via a text

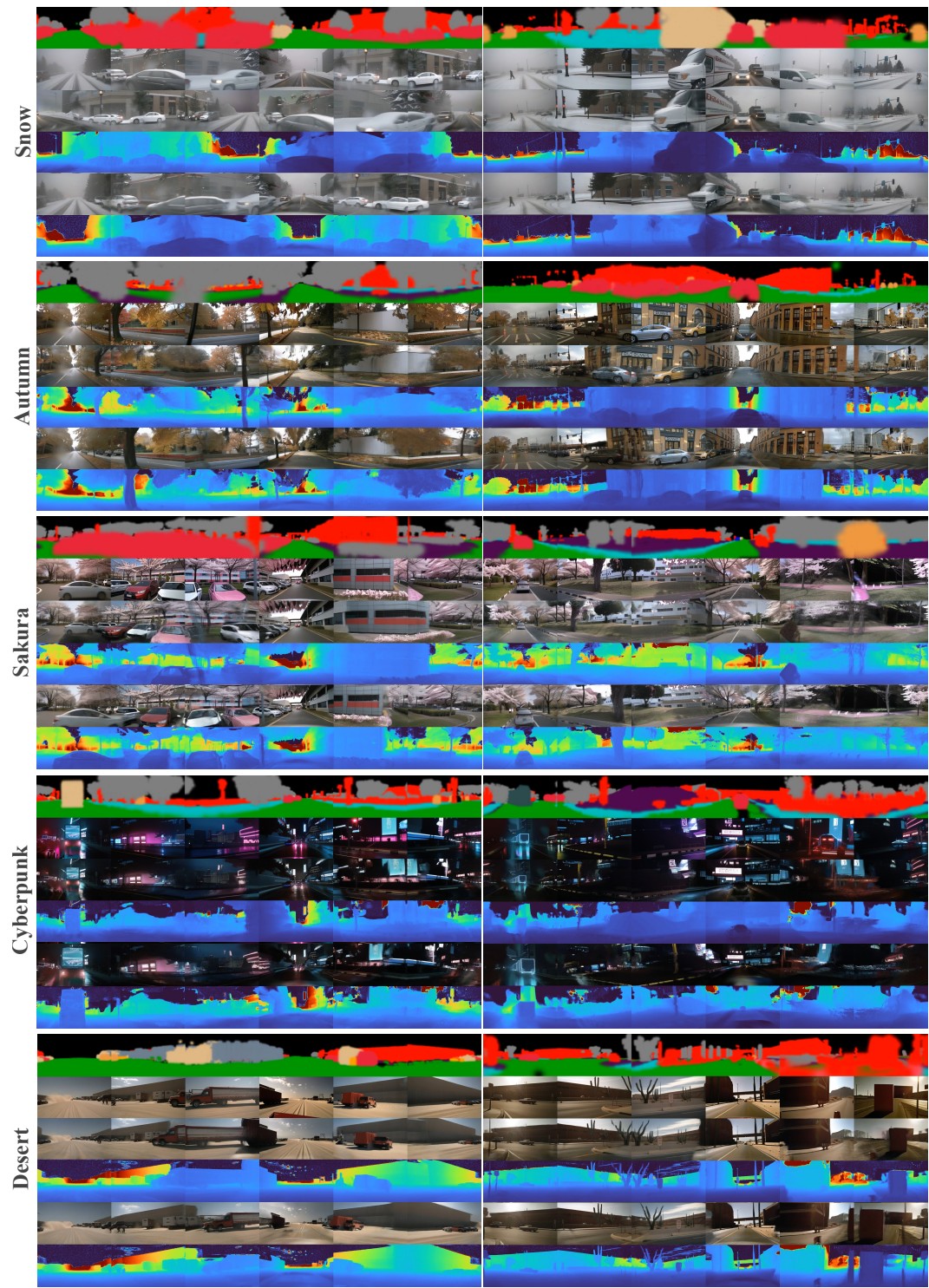

Figure 31: **Generalization Across Diverse Texture Domains.** We fine-tune SyntheOcc on SDXL to ensure superior style control.

prompt, while simultaneously dictating the precise spatial layout of scene elements (e.g., vehicles, roads) through the semantic map. As shown in Fig. 30 and Fig. 31, this dual-guidance mechanism yields highly controllable results that adhere to both textual and structural constraints.

### A.12.1 DETAILED ARCHITECTURE OF OUR MODEL.

Table 7 and Table 8 provide a detailed breakdown of our model's architecture, including the parameters for each perception branch, the configurations of the Cylinder Plane and CPFG modules, and key training hyperparameters. We trained our model on 8 NVIDIA L40S GPUs for approximately 2 days for occupancy prediction and 5 days for reconstruction. The training process required about 32 GB of GPU memory per device.

We employ Radio-V2.5 (Heinrich et al., 2024) as the 2D image encoder, which performs 16x downsampling to transform an input image of size 3x896x1600 (4x upsample to the original image) into a 768x56x100 feature map. These features are then used to construct a Cylinder Plane Feature by interpolating information from six source views into a 56x512 feature map, yielding a 768x56x512 feature volume with its Field of View (FOV) coefficients set to $\rho_o, \rho_v, \rho_p$ and height offsets set to $\Delta h_o, \Delta h_v, \Delta h_p$. For the volume-based module, the feature is processed by XNet, a U-Net-based network. Each encoder block contains two ResNet blocks and two ema attention modules, and each decoder block includes three corresponding blocks, with channel dimensions of 432, 624, and 624. XNet outputs two feature vectors of lengths 432 and 144, which are used for the construction of the Geometry CPFG and Appearance CPFG. Specifically, the 432-dimensional appearance feature is structured into an appearance CPFG of 48 equidistant cylindrical bins, each with a feature length of 36, while the 144-dimensional geometry feature is processed into a 12-dimensional Geometry CPFG. The YNet, which handles occlusion and pixel-level information, shares a similar architecture but with different feature dimensions as detailed in Table 7 and Table 8. The ZNet comprises three ResNet blocks with an upsample module followed by synthesis blocks with progressively decreasing feature lengths and finally synthesizing an RGB image. Each network has its corresponding decoders to form occupancy, BEV, geometry, and texture. We formulate occupancy prediction as a binary classification task to distinguish between occupied and free space, rather than predicting semantic labels. The module is trained with a composite loss $L_{occ}$ from Eq. (26), combining semantic ($L_{scal}^{sem}$), geometric ($L_{scal}^{geo}$), and cross-entropy ($L_{ce}$) losses from MonoScene (Cao & De Charette, 2022) with a BEV loss ($L_{bev}$) from FastOcc (Hou et al., 2024). The ground-truth labels $\mathbf{P}_{3D}$ and $\mathbf{P}_{2D}$ are generated as described in Appendix A.8. The terms $\lambda_{sem}, \lambda_{geo}, \lambda_{ce}, \lambda_{bev}$ are weights for each loss component.

$$
\begin{aligned}
L_{occ} = \lambda_{sem} L_{scal}^{sem}(\hat{\mathbf{P}}_{3D}, \mathbf{P}_{3D}) + \lambda_{geo} L_{scal}^{geo}(\hat{\mathbf{P}}_{3D}, \mathbf{P}_{3D}) \\
+ \lambda_{ce} L_{ce}(\hat{\mathbf{P}}_{3D}, \mathbf{P}_{3D}) + \lambda_{bev} L_{bev}(\hat{\mathbf{P}}_{2D}, \mathbf{P}_{2D})
\end{aligned}
\tag{26}
$$

The overall optimization objective of our model is composed of several loss components: an image reconstruction loss $L_1^I(\hat{\mathbf{I}}, \mathbf{I})$, a perceptual similarity loss $L_{lpips}(\hat{\mathbf{I}}, \mathbf{I})$, an L1 loss for the sky mask $L_{sky}(\hat{\mathbf{A}}_{fg}, \mathbf{A}_{fg})$, and both L1 and Pearson depth losses refer to (Xiong et al., 2023) for the depth map $L_1^{depth}(\hat{\mathbf{D}}_{fg}, \mathbf{D}_{fg}), L_{pear}(\hat{\mathbf{D}}_{fg}, \mathbf{D}_{fg})$, as formulated in Eq. (27). The ground truth for the depth map is generated by the Metric3D-v2 (Hu et al., 2024), while that for the sky mask is obtained from the LISA (Lai et al., 2024) as described in Appendix A.8. The terms $\lambda_I, \lambda_{lpips}, \lambda_{sky}, \lambda_{depth}, \lambda_{pear}$ are weights for each loss component.

$$
L_{total} = \lambda_I L_1^I + \lambda_{lpips} L_{lpips} + \lambda_{sky} L_{sky} + \lambda_{depth} L_1^{depth} + \lambda_{pear} L_{pear}
\tag{27}
$$

Table 7: Model architecture and training specifications of our model on nuScenes.

**(a) Network Architecture**

| | | |
|---|---|---|
| 2D Image Encoder | backbone
Out resolution | Radio-v2.5 (Heinrich et al., 2024)
$768 \times 56 \times 100$ |
| Cylinder Plane | resolution $D_{feat} \times H_u \times W_u$
#$\rho_o, \rho_v, \rho_p, \Delta h_o, \Delta h_v, \Delta h_p$ | $768 \times 56 \times 512$
0.9, 0.98, 0.98, 0.0, 0.4, 0.4 |
| Occupancy-Aware


Occupancy-CPFG | # blocks per resolution
# downsample dims
# upsample dims
# out dims
# CPFG planes num $K$
# CPFG planes dims $D_{occ}$ | 2
384, 576, 576
576, 576, 384
384
48
8 |
| Volume-Aware




Volume-CPFG | # blocks per resolution
# downsample dims
# upsample dims
# texture out dims
# geometry out dims
# CPFG planes num $K$
# CPFG planes dims $D_{geo}, D_{app}$ | 2
432, 624, 624
624, 624, 432
432
144
48
12, 36 |
| Pixel-Aware | # Upsampling factor $k_i, k_o$
# Upsample output dim $D_{pix}$
# blocks per resolution
# downsample dims
# upsample dims
# out dims | 4,1
128
1
128, 256, 512, 512
512, 512, 256. 128
128 |
| Background | # block dims
# block dims after ray cast
# target resolution $H_t, W_t$ | 24, 24, 24
24, 12, 6, 3
224, 400 |
| Occ Decoder


BEV Decoder | # MLP layers
# MLP input dims
# MLP width
# MLP output dims
# MLP layers
# MLP input dims
# MLP width
# MLP output dims | 4
8
4
4
3
128
64
2 |
| Pixel Decoder | # groupnorm channel
# groupnorm groups
# MLP layers
# MLP input dims
# MLP width
# MLP output dims
# Gaussians per pixel $G_p$ | 128
32
1
128
None
14
1 |
| Volume Decoder | # MLP layers
# MLP input dims
# MLP width
# MLP output dims
# Gaussians per voxel $G_v$ | 3
48
96
42
3 |

**(b) Hyperparameters**

| | | |
|---|---|---|
| Loss Weights | # $\lambda_{sem}, \lambda_{geo}, \lambda_{ce}, \lambda_{bev}$
# $\lambda_1, \lambda_{lpips}, \lambda_{sky}, \lambda_{depth}, \lambda_{pear}$ | 0.02, 0.02, 0.1, 0.1
1.0, 0.05, 0.5, 0.01, 0.01 |
| Training Details | learning rate scheduler
# iterations
# learning rate
optimizer
# beta1, beta2
# weight decay
# warm-up
# gradient clip | Cosine
100,000
1e-4
AdamW (Loshchilov & Hutter, 2019)
0.9, 0.999
0.01
1000
1.0 |

Table 8: Model architecture and training specifications of our model on Carla-Centric.

**(a) Network Architecture**

| | | |
|---|---|---|
| 2D Image Encoder | backbone | Radio-v2.5 (Heinrich et al., 2024) |
| | Out resolution | $768 \times 56 \times 100$ |
| Cylinder Plane | resolution $D_{feat} \times H_u \times W_u$ | $768 \times 56 \times 512$ |
| | #$\rho_o, \rho_v, \rho_p, \Delta h_o, \Delta h_v, \Delta h_p$ | 0.9, 1.6, 1.6,0.0,0.0,0.0 |
| Volume-Aware | # blocks per resolution | 2 |
| | # downsample dims | 432, 624, 624 |
| | # upsample dims | 624, 624, 432 |
| | # texture out dims | 432 |
| | # geometry out dims | 144 |
| Volume-CPFG | # CPFG planes num | 48 |
| | # CPFG planes dims $D_{geo}, D_{app}$ | 12,36 |
| Occupancy-Aware | # blocks per resolution | 2 |
| | # downsample dims | 384, 576, 576 |
| | # upsample dims | 576, 576, 384 |
| | # out dims | 384 |
| Occupancy-CPFG | # CPFG planes num $K$ | 48 |
| | # CPFG planes dims $D_{occ}$ | 8 |
| Pixel-Aware | # Upsampling factor $k_i, k_o$ | 2,2 |
| | # Upsample output dim $D_{pix}$ | 24 |
| | # blocks per resolution | 2 |
| | # downsample dims | 48, 48, 48 |
| | # upsample dims | 48, 48, 48 |
| | # out dims | 48 |
| Background | # block dims | 24, 24, 24 |
| | # block dims after ray cast | 24, 12, 6, 3 |
| | # target resolution $H_t, W_t$ | 150,200 |
| Occ Decoder | # MLP layers | 4 |
| | # MLP input dims | 8 |
| | # MLP width | 4 |
| | # MLP output dims | 4 |
| BEV Decoder | # MLP layers | 3 |
| | # MLP input dims | 128 |
| | # MLP width | 64 |
| | # MLP output dims | 2 |
| Pixel Decoder | # groupnorm channel | 24 |
| | # groupnorm groups | 6 |
| | # MLP layers | 4 |
| | # MLP input dims | 24 |
| | # MLP width | 24 |
| | # MLP output dims | 14 |
| | # Gaussians per pixel $G_p$ | 1 |
| Volume Decoder | # MLP layers | 8 |
| | # tex MLP input dims | 48 |
| | # geo MLP input dims | 12 |
| | # MLP width | 64 |
| | # tex MLP output dims | 33 |
| | # geo MLP output dims | 9 |
| | # Gaussians per voxel $G_v$ | 3 |

**(b) Hyperparameters**

| | | |
|---|---|---|
| Loss Weights | # $\lambda_{sem}, \lambda_{geo}, \lambda_{ce}, \lambda_{bev}$ | 0.02, 0.02, 0.1, 0.1 |
| | # $\lambda_1, \lambda_{lpips}, \lambda_{sky}, \lambda_{depth}, \lambda_{pear}$ | 1.0, 0.05, 0.5, 0.01, 0.01 |
| Training Details | learning rate scheduler | Cosine |
| | # iterations | 100,000 |
| | # learning rate | 1e-4 |
| | optimizer | AdamW (Loshchilov & Hutter, 2019) |
| | # beta1, beta2 | 0.9, 0.999 |
| | # weight decay | 0.01 |
| | # warm-up | 1000 |
| | # gradient clip | 1.0 |

