# OpenReview forum: "XYZCylinder: Feedforward Reconstruction for Driving Scenes Based on A Unified Cylinder Lifting Method"
_ICLR.cc/2026/Conference — ICLR 2026 Conference Withdrawn Submission_

### Official Review · Reviewer_tf9T · 2025-10-23

**Soundness:** 2
**Presentation:** 2
**Contribution:** 2
**Rating:** 4
**Confidence:** 5

**Summary:**

Recent attention has focused on feedforward reconstruction paradigms, which implicitly learn fixed view transformations and reconstruct scenes with a single representation. However, these paradigms have limited generalization (due to fixed view transformations failing with changed camera configurations) and reconstruction accuracy (due to sparse panorama view overlaps and complex driving scenes) in driving scenarios. To address these, the authors propose XYZCylinder, a feedforward model with a unified cylinder lifting method (including camera modeling and feature lifting). Specifically, it uses Unified Cylinder Camera Modeling (UCCM) to improve generalization by unifying camera configurations, and a hybrid representation with Cylinder Plane Feature Group (CPFG) to boost accuracy by lifting 2D features to 3D. Experiments show it achieves SOTA performance and zero-shot generalization to other driving scenes.

**Strengths:**

1. A large number of methods were compared on multiple datasets.
2. The methods were clearly expressed.
3. The formulas and diagrams were clearly defined and presented.

**Weaknesses:**

This feedback raises four key criticisms and suggestions for the paper, focusing on **model complexity, rendering limitations, missing efficiency metrics, and mismatched advantages for target applications**:

1. **Model & Realism Issues**: The model’s design is overcomplicated, and its rendering realism lags behind optimization-based methods (e.g., 3DGS). It also only supports low resolution and a limited number of input images.
2. **Practical Applicability Gap**: The low rendering quality fails to meet practical needs like camera simulation; additionally, the model relies on depth map ground truth for training—unlike competitors (Pixelsplat, MVSpalt) that avoid this, restricting its real-world use.
3. **Missing Efficiency Analysis**: The paper claims a feed-forward approach but lacks critical metrics like inference latency, running time, and memory consumption. Competitors (MVSplat, PixelSplat) include these to prove efficiency, so this analysis is needed for fair comparison.
4. **Mismatched Advantages for Target Use Case**: For the stated application (autonomous driving real-world simulators), offline scene-optimized 4D reconstruction is already acceptable (no high efficiency demands). Meanwhile, the feed-forward method’s key strength (generalization) does not offset its lower reconstruction accuracy compared to scene-optimized approaches.

**Questions:**

The problems are all listed above. Moreover, the scope for innovation in methods is limited.

---

### Official Review · Reviewer_snvb · 2025-10-29

**Soundness:** 2
**Presentation:** 1
**Contribution:** 1
**Rating:** 2
**Confidence:** 3

**Summary:**

This paper proposes XYZCylinder, a feedforward 3D reconstruction framework for driving scenes based on a Unified Cylinder Lifting Method (UCCM). The system replaces learned view transformations with an explicit cylindrical camera model and introduces a hybrid architecture consisting of three main modules: an occupancy-aware network to prune empty voxels, a volume-aware network to generate Gaussian primitives, and a pixel-aware network for texture refinement.

The authors claim that this design improves both reconstruction accuracy and generalization across different camera configurations. Experiments are conducted on nuScenes and Carla-Centric datasets, with qualitative zero-shot evaluations on other driving datasets.

**Strengths:**

1. The proposed UCCM is an interesting attempt to unify camera geometry across multi-view driving data.

2. The experimental results show consistent improvements over existing feedforward 3DGS baselines (e.g., Omni-Scene, DrivingForward) on nuScenes and Carla-Centric.

**Weaknesses:**

1. **Too much hand-crafted complexity.**
The overall architecture feels over-engineered. For instance, the clockwise/counter-clockwise flipping and feature-fusion design adds a lot of manual logic but lacks evidence that it’s actually necessary. Simple feature averaging or learned fusion might achieve similar results with much simpler design. The paper would benefit from clearer ablations to justify these choices.
2. **Occupancy module is an engineering trick.**
The occupancy-aware network seems to exist mostly to reduce GPU memory use, not as a conceptual innovation. The authors do not quantify how much memory it saves or how it scales, which makes the contribution feel more like a system workaround than a core method.
3. **Foreground/background decoupling is awkward.**
The method treats the foreground (modeled by 3DGS) and the background (generated by a RGB decoder) separately, then fuses them for rendering. This design makes the pipeline incompatible with other 3DGS methods that produce a unified scene representation.
If the goal is actually novel-view synthesis, the paper should compare against works like ViewCrafter[1] or GEN-3C[2], which also synthesize images directly rather than reconstruct explicit 3DGS. As it stands, the comparison set does not match the claimed task.
And it’s never explained how the system decides what counts as foreground versus background, through semantic masks? depth thresholding? heuristics? This missing detail is important, since the whole pipeline depends on it.
4. **Poor Evaluation**
The central claim is zero-shot generalization across datasets, but this is only shown through qualitative figures, not quantitative evaluation. Those results should have been the main experiment, yet the paper focuses mostly on in-domain reconstruction metrics (PSNR/SSIM). Without solid cross-dataset numbers, the generalization claim is weak.
5. **Conceptual novelty is overstated.**
Much of the system relies on pre-trained models (Radio-v2.5, Metric3D-v2) and heuristic tuning of geometric parameters (ρ, ∆h). The main novelty is a deterministic projection mapping, is presented as an improvement over learned projections, yet no direct comparison is provided to support this claim.

---

[1] Yu, W., Xing, J., Yuan, L., Hu, W., Li, X., Huang, Z., Gao, X., Wong, T.-T., Shan, Y., & Tian, Y. (2025). ViewCrafter: Taming video diffusion models for high-fidelity novel view synthesis. IEEE Transactions on Pattern Analysis and Machine Intelligence.
[2] Ren, X., Shen, T., Huang, J., Ling, H., Lu, Y., Nimier-David, M., Müller, T., Keller, A., Fidler, S., & Gao, J. (2025). GEN3C: 3D-Informed world-consistent video generation with precise camera control. Proceedings of the IEEE/CVF Conference on Computer Vision and Pattern Recognition (CVPR).

**Questions:**

The paper is poorly written. Although it presents a substantial amount of work, much of it feels confusing, poorly organized, and driven by heuristics rather than by clear necessity. The authors should consider rewriting the paper with clearer motivation, better organization, and a more coherent overall flow.

---

### Official Review · Reviewer_J1JH · 2025-10-31

**Soundness:** 3
**Presentation:** 3
**Contribution:** 3
**Rating:** 4
**Confidence:** 4

**Summary:**

This paper proposes XYZCylinder, a feed-forward 3D scene reconstruction pipeline for surround-view driving data. The core idea is a Unified Cylinder Camera Modeling (UCCM) that projects all camera views to a cylindrical plane with shared parameters (center (c_u^x), radius (R_u), height (Z_u), resolution). Two composited cylinder-plane features (F^+{cy}) and (F^-{cy}) are built by overlaying views in clockwise and counter-clockwise orders using an ordered composition operator (\gamma) (overwrite-on-nonzero). Foreground is reconstructed with three branches (occupancy/volume/pixel) that lift features into CPFGs; background is rendered separately and then fused. The model uses strong 2D backbones (Radio-v2.5) and depth cues (Metric3D-v2). On nuScenes/Carla-Centric, the paper reports modest SOTA improvements (e.g., PSNR 24.97 vs. 24.11 for Omni-Scene on nuScenes).

**Strengths:**

1. The cylinder center is the mean camera position with a small height offset; effective vertical FOV is set to the dataset’s minimum (scaled by (\rho)) to avoid top/bottom holes during projection; the radius is derived analytically. This is simple and likely robust across rigs.
2. Foreground/background decoupling and the occupancy-aware gating are sensible; the ablation shows that removing occupancy causes OOM and hurts metrics, supporting the design.
3.  The paper explicitly lists inter-dataset camera differences and tunes UCCM parameters per dataset; for Waymo it introduces symmetric virtual cameras.

**Weaknesses:**

1. The proposed method integrates numerous modules, yet the overall gains—especially on the nuScenes dataset—are modest. Coupled with the lack of complexity and runtime comparisons (e.g., parameters, FLOPs, latency), the work appears to require substantial engineering effort for limited benefit.
2. The experimental results do not substantiate the stated motivation. While the Introduction emphasizes generalization, the observed generalization is very likely driven by the powerful base models rather than by the contribution of the proposed method itself.
3. The ordered composition is formally defined, and the projection math in the appendices appears correct. However, without runtime numbers and without ablations on the composition operator, it is hard to judge practical efficiency and robustness.
4. Several implementation choices (∆h/ρ rationale; details of how the two traversal orders influence occlusion disambiguation) should be better motivated.

**Questions:**

1.	Why introduce a small height offset (Δh)? What is its role?

2.	Why does using the minimum vertical FoV alleviate multi-camera mismatch? What is the underlying mechanism?

3.	What is the rationale for “overwriting the existing ones with the incoming ones” during composition? Have you compared against averaging, max pooling, confidence-/depth-weighted blending, etc.?

4.	How do the last clockwise/counter-clockwise views—i.e., the two composites F_cy^+ and F_cy^−—together with YNet_occ help distinguish occupied vs. empty space?

5.	Efficiency reporting is incomplete. The appendix lists training resources (8×L40S; ≈2 days for occupancy, ≈5 days for reconstruction; ~32 GB per GPU) but not parameter counts, FLOPs, end-to-end latency, or FPS. Please provide: total parameters; training/inference FLOPs; end-to-end per-frame latency (including feature extraction, projection, the three branches, and rendering); peak memory; and a speed–quality comparison against PixelSplat/MVSplat/DrivingForward/Omni-Scene.

6.	Backbone and external depth dependence. You use Radio-v2.5 as the 2D backbone and Metric3D-v2 for depth/confidence to produce depth-augmented F̄_cy^±. Please report performance and speed with lighter backbones and with no external depth, to isolate the contribution of UCCM/CPFG.

7.	Zero-shot attribution. It remains unclear whether the zero-shot gains come primarily from the backbone rather than the proposed method. Please clarify with appropriate controls/ablations.

---

### Official Review · Reviewer_owgh · 2025-10-31

**Soundness:** 3
**Presentation:** 3
**Contribution:** 3
**Rating:** 6
**Confidence:** 4

**Summary:**

The paper “XYZCylinder: Feedforward Reconstruction for Driving Scenes Based on a Unified Cylinder Lifting Method” proposes a novel feedforward 3D reconstruction framework tailored for autonomous driving. Unlike conventional NeRF-style iterative optimization or fixed-view feedforward models, XYZCylinder introduces a Unified Cylinder Camera Modeling (UCCM) strategy that explicitly maps multi-view inputs into a shared cylindrical coordinate system, enabling zero-shot generalization across different camera configurations. The model employs a hybrid representation consisting of occupancy-, volume-, and pixel-aware modules organized via a Cylinder Plane Feature Group (CPFG), allowing fine-grained geometry and texture reconstruction. Overall, I like the motivations and contributions of this work.

**Strengths:**

+ Novel Unified Cylinder Modeling. The proposed UCCM elegantly handles heterogeneous multi-camera configurations, offering an explicit, geometry-consistent mapping that enables cross-dataset and zero-shot reconstruction — a significant step beyond view-specific feedforward models.

+ Hybrid 3D Representation Design. The combination of occupancy-, volume-, and pixel-aware modules yields a balanced trade-off between geometric completeness and photometric fidelity, outperforming both volumetric and pixel-based baselines.

+ Strong Empirical Results and Analysis. The paper provides comprehensive quantitative and qualitative evaluations, including detailed ablations on module contributions and network design (e.g., dual-branch encoders), establishing clear evidence for the claimed improvements and generalization ability.

**Weaknesses:**

- While the architecture achieves high performance, the model’s modular and multi-branch design (YNet, XNet, ZNet) is intricate, which might hinder reproducibility and limit theoretical interpretability of the learned representations.
- The cylinder-style perspective has been widely explored in the 360 panoramic field. For example, "Eliminating the Blind Spot: Adapting 3D Object Detection and Monocular Depth Estimation to 360◦ Panoramic Imagery" adapts contemporary deep network architectures developed on conventional rectilinear imagery, "Cylin-Painting: Seamless 360° Panoramic Image Outpainting and Beyond" and "PanoDiffusion: 360-degree Panorama Outpainting via Diffusion" use a cylinder-like convolution or circular padding to ensure the seamless content generation of panoramas. The brief discussions of these works are also expected to be presented in this work.
- Could the proposed hybrid representation be extended beyond driving scenes (e.g., indoor or aerial settings), or are there inherent assumptions tied to the cylindrical geometry of street-view cameras?

**Questions:**

The unified cylinder projection depends on adjustable parameters. How sensitive is the model performance to these hyperparameters, and can they be automatically inferred or adapted during training?

---

### Note · Authors · 2025-11-13

I have read and agree with the venue's withdrawal policy on behalf of myself and my co-authors.